# Tackling Continual Offline RL through Selective Weights Activation on Aligned Spaces

**Jifeng Hu**[1]  **Sili Huang**[2*]  **Li Shen**[3]  **Zhejian Yang**[1]  **Shengchao Hu**[4]
**Shisong Tang**[5]  **Hechang Chen**[1*]  **Lichao Sun**[6]
**Yi Chang**[1*]  **Dacheng Tao**[7]

[1]Jilin University  [2]Minzu University of China  [3]Shenzhen Campus of Sun Yat-sen University
[4]Shanghai Jiao Tong University  [5]Tsinghua University  [6]Lehigh University
[7]Nanyang Technological University

{hujf21, zjyang22}@mails.jlu.edu.cn {chenhc, yichang}@jlu.edu.cn
huangsili@muc.edu.cn mathshenli@gmail.com charles-hu@sjtu.edu.cn
tangshisong13@gmail.com lis221@lehigh.edu dacheng.tao@gmail.com

## Abstract

Continual offline reinforcement learning (CORL) has shown impressive ability in diffusion-based continual learning systems by modeling the joint distributions of trajectories. However, most research only focuses on limited continual task settings where the tasks have the same observation and action space, which deviates from the realistic demands of training agents in various environments. In view of this, we propose Vector-Quantized Continual Diffuser, named VQ-CD, to break the barrier of different spaces between various tasks. Specifically, our method contains two complementary sections, where the quantization spaces alignment provides a unified basis for the selective weights activation. In the quantized spaces alignment, we leverage vector quantization to align the different state and action spaces of various tasks, facilitating continual training in the same space. Then, we propose to leverage a unified diffusion model attached by the inverse dynamic model to master all tasks by selectively activating different weights according to the task-related sparse masks. Finally, we conduct extensive experiments on 15 continual learning (CL) tasks, including conventional CL task settings (identical state and action spaces) and general CL task settings (various state and action spaces). Compared with 17 baselines, our method reaches the SOTA performance.

## 1 Introduction

The endeavor of recovering high-performance policies from abundant offline samples gathered by various sources and continually mastering future tasks learning and previous knowledge maintaining gives birth to the issue of continual offline reinforcement learning (CORL) [62, 2, 39, 42]. Ever-growing scenarios or offline datasets pose challenges for most continual RL methods that are trained on static data and are prone to showing catastrophic forgetting of previous knowledge and ineffective learning of new tasks [63, 57]. Facing these challenges, three categories of methods, rehearsal-based [42, 74, 10], regularization-based [82, 105, 104], and structure-based methods [102, 67, 8], are proposed to reduce forgetting and facilitate continual training.

However, most previous studies only focus on the continual learning (CL) setting with identical state and action spaces [63, 82]. It deviates from the fact that the ever-growing scenarios or offline datasets are likely to possess different state and action spaces with previous tasks for many reasons,

---

*Corresponding authors: Hechang Chen, Sili Huang, and Yi Chang.

39th Conference on Neural Information Processing Systems (NeurIPS 2025).

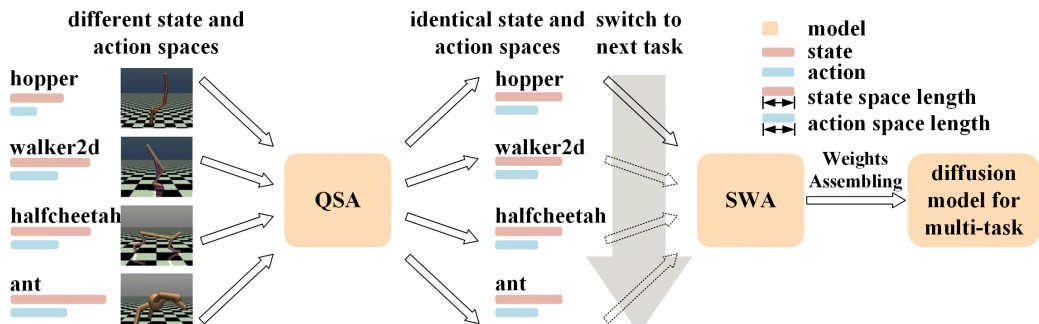

Figure 1: The high-level intuition of VQ-CD. Quantized space alignment (QSA) is used to expand the application range of VQ-CD, while selective weight activation (SWA) is used to reduce forgetting of historical tasks in continual learning.

such as the variation of demands and the number of sensors [98, 102]. Moreover, these datasets often come from multiple behavior policies, which pose the additional challenge of modeling the multimodal distribution of various tasks [2, 60]. Benefiting from diffusion models' powerful expressive capabilities and highly competitive performance, an increasing number of researchers are considering incorporating them to address the CORL problems [4, 99, 17] from the perspective of sequential modeling. There have been several attempts to combine diffusion-based models with rehearsal-based and regularization-based techniques, which usually apply constraints to the continual model learning process with previous tasks' data or well-trained weights [82, 99, 63]. However, constrained weight updating will limit the learning capability of new tasks and can not preserve the previously acquired knowledge perfectly [98]. Although structure-based methods can eliminate forgetting and strengthen the learning capability by preserving well-trained weights of previous tasks and reserving disengaged weights for ongoing tasks, they are still limited in simple architectures and CL settings with identical state and action spaces [102, 93, 66]. Thus, in this paper, we seek to answer the question:

*Can we merge the merits of diffusion models' powerful expression and structure-based parameters isolation to master CORL problems with any task sequence?*

We answer this in the affirmative through the key insight of allocating harmonious weights for each continual learning task. Figure 1 shows the intuitive design of our methods. Specifically, we propose Vector-Quantized Continual Diffuser called VQ-CD, which contains two complementary sections: the quantized spaces alignment (QSA) module and the selective weights activation diffuser (SWA) module. To expand our method to any task sequences under the continual learning setting, we adopt the QSA module to align the different state and action spaces. Concretely, we adopt vector quantization to map the task spaces to a unified space for training based on the contained codebook and recover it to the original task spaces for evaluation. In the SWA module, we first perform task mask generation for each task, where the task masks are applied to the one-dimensional convolution kernel of the U-net structure diffusion model. Then, we use the masked kernels to block the influence of unrelated weights during the training and inference. Finally, after the training process, we propose the weights assembling to aggregate the task-related weights together for simplicity and efficiency. In summary, our main contributions are fourfold:

- We propose the Vector-Quantized Continual Diffuser (VQ-CD) framework, which can not only be applied to conventional continual tasks but also be suitable for any continual tasks setting, which makes it observably different from the previous CL method.
- In the quantized spaces alignment (QSA) module of VQ-CD, we adopt ensemble vector quantized encoders based on the constrained codebook because it can be expanded expediently. During the inference, we apply task-related decoders to recover the various observation and action spaces.
- In the selective weights activation (SWA) diffuser module of VQ-CD, we first perform task-related task masks, which will then be used to the kernel weights of the diffuser. After training, we propose assembling weights to merge all learned knowledge.
- Finally, we conduct extensive experiments on 15 CL tasks, including conventional CL settings and any CL task sequence settings. The results show that our method surpasses or matches the SOTA performance compared with 17 representative baselines.

## 2 Related Work

**Offline RL.**  Offline reinforcement learning is becoming an important direction in RL because it supports learning on large pre-collected datasets and avoids massive demand for expensive, risky interactions with the environments [69, 71, 34, 7, 31, 46, 45]. Directly applying conventional RL methods in offline RL faces the challenge of distributional shift [78, 62, 96, 100, 44] caused by the mismatch between the learned and data-collected policies, which will usually make the agent improperly estimate expectation return on out-of-distribution actions [78, 58, 2, 43, 47]. To tackle this challenge, previous studies try to avoid the influences of out-of-distribution actions by adopting constrained policy optimization [27, 20, 71, 58], behavior regularization [70, 59, 23, 37], importance sampling [49, 24, 103], uncertainty estimation [3, 89, 61], and imitation learning [94, 81, 13, 40].

**Continual RL.**  Continual learning (CL) aims to solve the plasticity and stability trade-off under the task setting, where the agent can only learn to solve each task successively [92]. CL can be classified into task-aware CL and task-free CL according to whether there are explicit task boundaries [5, 92]. In this paper, we mainly focus on task-aware CL. There are three main technical routes to facilitate forward transfer (plasticity) and mitigate catastrophic forgetting (stability). Rehearsal-based approaches [80, 66, 93, 102, 82] store a portion of samples from previous tasks and use interleaving updates between new tasks' samples and previous tasks' samples. Simply storing samples increases the memory burden in many scenarios; thus, generative models such as diffusion models are introduced to mimic previous data distribution and generate synthetic replay for knowledge maintenance [101, 75, 22]. Regularization-based approaches [51, 52, 104, 105] seek to find a proficiency compromise between previous and new tasks by leveraging constraint terms on the total loss function. Usually, additional terms of learning objectives will be adopted to penalize significant changes in the behaviors of models' outputs or the updating of models' parameters [55, 51]. In the structure-based approaches [93, 53, 93, 102, 82, 56], researchers usually consider parameter isolation by using sub-networks or task-related neurons to prevent forgetting.

**Diffusion RL.**  Recently, diffusion-based models have shown huge potential in RL under the perspective of sequential modeling [30, 32, 48, 4, 38, 36]. A typical use of diffusion models is to mimic the joint distribution of states and actions, and we usually use state-action value functions as the classifier or class-free guidance when generating decisions [72, 29, 73, 33, 41]. Diffusion models, as representative generative models, can also be used as environmental dynamics to model and generate synthetic samples to improve sample efficiency or maintain previous knowledge in CL [97, 28, 65, 16, 63]. It is noted that the diffusion model's powerful expression ability on multimodal distribution also makes it suitable for being used as policies to model the distribution of actions and as planners to perform long-horizon planning [91, 50, 11]. Besides, diffusion models can also be used as multi-task learning models to master several tasks simultaneously [26] or as multi-agent models to solve more complex RL scenarios [106].

## 3 Preliminary

### 3.1 Continual Offline RL

We focus on the task-aware CL in the continual offline RL in this paper [1, 92, 82, 76, 90]. Suppose that we have $I$ successive tasks, and task $j$ arises behind task $i$ for any $i < j$. Each task $i, i \in [1 : I]$ is represented by a Markov Decision Process (MDP) $\mathcal{M}^i = \langle \mathcal{S}^i, \mathcal{A}^i, \mathcal{P}^i, \mathcal{R}^i, \gamma \rangle$, where we use superscript $i$ to differentiate different tasks, $I$ is the number of total tasks, $\mathcal{S}$ is the state space, $\mathcal{A}$ is the action space, respectively, $\mathcal{P} : \mathcal{S} \times \mathcal{A} \to \Delta(\mathcal{S})$ denotes the transition function, $\mathcal{R} : \mathcal{S} \times \mathcal{A} \times \mathcal{S} \to \mathbb{R}$ is the reward function, and $\gamma \in [0, 1)$ is the discount factor. Conventional CL tasks have the same state and action spaces for all tasks, i.e., $|\mathcal{S}^i| = |\mathcal{S}^j|, |\mathcal{A}^i| = |\mathcal{A}^j|, \forall i, j \in [1 : I]$. While for any tasks sequences, we have $|\mathcal{S}^i| \neq |\mathcal{S}^j|, |\mathcal{A}^i| \neq |\mathcal{A}^j|$. In the offline RL, we can only access pre-collected datasets $\{D^i\}_{i \in [1:I]}$ of each task. The goal of continual offline RL is to find an optimal policy that can maximize the objective $\sum_i^I \mathbb{E}_\pi [\sum_{t=0}^{\infty} \gamma^t r(s_t^i, a_t^i)]$ [19, 98, 84] on all tasks.

### 3.2 Conditional Generative Behavior Modeling

In this paper, we adopt the diffusion-based model with the U-net backbone as the generative model to fit the joint distribution $q(\tau_s) = \int q(\tau_s^{0:K}) d\tau_s^{1:K}$ of state sequences $\tau_s$ and an inverse dynamics

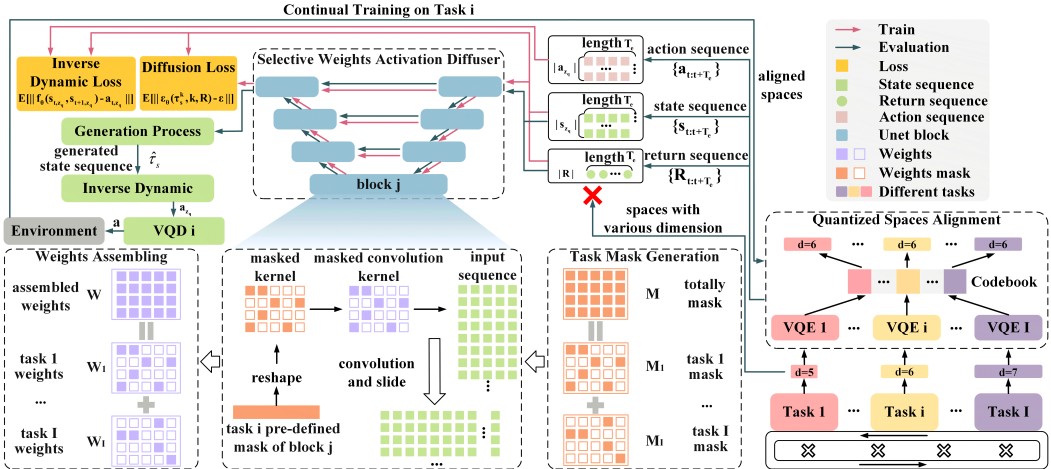

Figure 2: The framework of VQ-CD. It contains two sections: The Quantized Space Alignment (QSA) module and the Selective Weights Activation (SWA) module, where QSA enables our method to adapt to general task-aware continual learning task settings by transferring the different state and action spaces to the same spaces. SWA uses selective neural network weight activation to maintain the knowledge of previous tasks through task-related weight masks. After the training, we perform Weights Assembling to integrate the total weights and save the memory budget.

model $f_{inv,\psi}(s_t, s_{t+1})$ to produce actions $a_t$, where $k \in [1 : K]$ is the diffusion step, $t$ is the RL time step, $\psi$ is the parameters of inverse dynamics model, and we omit the identification of tasks for the sake of simplicity because the training is same for all tasks. Through specifying the pre-defined forward diffusion process $q(\tau_s^k|\tau_s^{k-1}) = \mathcal{N}(\tau_s^k; \sqrt{\alpha_k}\tau_s^{k-1}, \beta_k \boldsymbol{I})$ and the trainable reverse process $p_\theta(\tau_s^{k-1}|\tau_s^k) = \mathcal{N}(\tau_s^{k-1}; \mu_\theta(\tau_s^k, k), \Sigma^k)$ [30], we can train the diffusion model with the simplified loss function

$$\mathcal{L}(\theta) = \mathbb{E}_{k \sim U(1,2,...,K), \epsilon \sim \mathcal{N}(0,\boldsymbol{I}), \tau_s^0 \sim D, b \sim \mathcal{B}(\lambda)}[||\epsilon - \epsilon_\theta(\tau_s^k, k, b * \mathcal{C})||_2^2], \quad (1)$$

where $\tau_s^k = \sqrt{\bar{\alpha}_k}\tau_s^0 + \sqrt{1 - \bar{\alpha}_k}\epsilon$, $\mu_\theta(\tau_s^k) = \frac{1}{\sqrt{\alpha_k}}(\tau_s^k - \frac{\beta_k}{\sqrt{1-\bar{\alpha}_k}}\epsilon_\theta(\tau_s^k, k))$, $\Sigma^k = \frac{1-\bar{\alpha}_{k-1}}{1-\bar{\alpha}_k}\beta_k \boldsymbol{I}$, $\alpha_k$ is the approximate discretization pre-defined parameters [12, 64], $\beta_k = 1 - \alpha_k$, $\bar{\alpha}_k = \prod_{\iota=1}^k \alpha_\iota$, $U$ is the uniform distribution, $\epsilon$ is standard Gaussian noise, $\boldsymbol{I}$ is the identity matrix, $\tau_s^0 \sim D$ is the state sequences stored in the task replay buffer $D$, $\mathcal{B}$ is binomial distribution, $\lambda = 0.25$ is the parameter of $\mathcal{B}$, $\mathcal{C}$ is condition, which is usually selected as discounted returns or value function in RL, and $\theta$ is the total parameters of model $\epsilon_\theta$. The following is

$$\hat{\tau}_s^{k-1} = \frac{1}{\sqrt{\alpha_k}}(\hat{\tau}_s^k - \frac{\beta_k}{\sqrt{1-\bar{\alpha}_k}}\hat{\epsilon}) + \sqrt{\frac{1-\bar{\alpha}_{k-1}}{1-\bar{\alpha}_k}\beta_k}\epsilon. \quad (2)$$

generation function, where we use $\hat{\tau}_s^k$ to denote the generated state sequences, $\hat{\epsilon} = \epsilon_\theta(\hat{\tau}_s^k, k, \emptyset) + \omega(\epsilon_\theta(\hat{\tau}_s^k, k, \mathcal{C}) - \epsilon_\theta(\hat{\tau}_s^k, k, \emptyset))$, $\omega$ is the guidance scale, $\emptyset$ means $b = 0$. We use inverse dynamics model $f_{inv,\psi}(\cdot)$ to produce actions, where the training loss is

$$\mathcal{L}(\psi) = \mathbb{E}_{(s_t, a_t, s_{t+1}) \sim D}[||a_t - f_{inv,\psi}(s_t, s_{t+1})||_2^2]. \quad (3)$$

## 4 Method

Our method enables training on general task-aware CL task sequences through two sections (as shown in Figure 2): the selective weights activation diffuser (SWA) module and the quantized spaces alignment (QSA) module. Algorithm 2 shows how to generate the actions during inference. The detailed training process is shown in Algorithm 1 of Appendix A.1. In the following parts, we introduce these two modules in detail.

Table 1: The comparison of VQ-CD, diffusion-based baselines, and LoRA methods on Ant-dir tasks, where the continual task sequence is 10-15-19-25. The results are average on 30 evaluation rollouts with 30 random seeds.

| Method | VQ-CD (ours) | CoD | Multitask CoD | IL-rehearsal | CoD-LoRA | Diffuser-w/o rehearsal | CoD-RCR | MTDIFF | DD-w/o rehearsal |
|---|---|---|---|---|---|---|---|---|---|
| Mean return | $558.22_{\pm 1.14}$ | $478.19_{\pm 15.84}$ | $485.15_{\pm 5.86}$ | $402.53_{\pm 17.67}$ | $296.03_{\pm 11.95}$ | $270.44_{\pm 5.54}$ | $140.44_{\pm 32.11}$ | $84.01_{\pm 41.10}$ | $-11.15_{\pm 45.27}$ |

## 4.1 Quantized Spaces Alignment

To make our method suitable for solving any CL task sequence setting, we propose aligning the different state and action spaces with the quantization technique. Specifically, we propose to solve the following quantized representation learning problem

$$\min_{\theta_e, \theta_d, \theta_q} \quad \mathcal{L}_{QSA}(x; \theta_e, \theta_d, \theta_q),$$
$$\text{s.t.} \quad ||z_q||_2^2 < \rho, \tag{4}$$

where $\mathcal{L}_{QSA}(x) = \mathbb{E}\left[||x - f_{VQD}(z_q; \theta_d)||_2^2\right] + \mathbb{E}\left[||\mathbf{sg}(z_q) - z_e||_2^2\right] + \mathbb{E}\left[||\mathbf{sg}(z_e) - z_q||_2^2\right]$ is the total quantized loss, $\mathbf{sg}(\cdot)$ represents the stop gradient operation, $\theta_e$ and $\theta_d$ are the parameters of the vector quantized encoder (VQE) and vector quantized decoder (VQD), $\theta_q$ is the parameters of the codebook, $\rho$ limits the range of codebook embeddings, $x$ can represent the states or actions for each specific CL task, $z_q = f_{\theta_q}(z_e)$ is the quantized representation which is consisted of fixed number of fixed-length quantized vectors, and $z_e = f_{VQE}(x; \theta_e)$ is the output of the encoder. Here, we propose searching the constrained optimal solution of the above problem for the consideration of the diffusion model training within a limited value range, just like the limit normalization in CV [30, 14] and RL [4, 64]. There are many methods to force optimization under restricted constraints, such as converting the constraints to a penalty term [9]. In our method, for simplicity and convenience, we propose to directly clip the quantized vector $z_q$ to meet the constraints after every codebook updating step. Moreover, to meet the potential demand for extra tasks beyond the predefined CL tasks, we design the codebook as easy to equip, where the quantized spaces of different tasks are separated so that we can expediently train new task-related encoders, decoders, and quantized vectors.

For tasks where the state and action spaces are different, we can use the well-trained QSA module to obtain the aligned state feature $s^i_{z_q} = f^i_{s, \theta_q}(f^i_{VQE_s}(s^i; \theta_e))$ and the action feature $a^i_{z_q} = f^i_{a, \theta_q}(f^i_{VQE_a}(a^i; \theta_e))$ for each task $i$. Thus, we can use $\tau^i_{s_{z_q}}$ and $\tau^i_{a_{z_q}}$ to represent the state and action feature sequences. Now, the action is produced through $a^i_t = f^i_{VQD_a}(f_{inv}(s_{z_q,t}, s_{z_q,t+1}); \theta_d)$.

## 4.2 Selective Weights Activation

In this section, we introduce how to selectively activate different parameters of the diffusion model to reduce catastrophic forgetting and reserve disengaged weights for ongoing tasks.

**Task Mask Generation.** Suppose that the diffusion model contains $L$ blocks, and the weights (i.e., parameters) of block $l$ are denoted by $W_l, l \in \{1, ..., L\}$. There are two ways to disable the influence of the weights on the model outputs. One is masking the output neurons $O_l = f_l(\cdot; W_l)$ of each block, where $f_l(\cdot)$ is the neural network function of block $l$. This strategy is friendly to MLP-based neural networks for two reasons: 1) the matrix calculation, such as $W_l * x$, is relatively simple so that we can easily recognize the disabled weights; 2) we do not need to apply any special operation on the optimizer because the output masking will cut off gradient flow naturally. However, we can not arbitrarily apply the above masking strategy to more expressive network structures, such as convolution-based networks, because we can not easily distinguish the dependency between parameters and outputs. Thus, we search for another masking strategy: masking the parameters $W_l$ with $M_l$, which permits us to control each parameter accurately.

Specifically, suppose that the total available mask positions of block $l$ are $M_l$. In this paper, $M_l$ is a ones matrix, and the entries with 0 mean that we will perform masking. Before training on task $i$, we first pre-define the specific mask $M_{i,l}$ of task $i$ on block $l$ by randomly sampling unmasked positions from the remaining available mask positions. Then, with the increase of the tasks, the remaining available mask positions decrease until $M_l = \sum_{i=1}^{I} M_{i,l}$.

**Selective Weights Forward and Backward Propagation.** After obtaining the mask $M_{i,l}$, we can perform forward propagation with masked weights

$$
\begin{aligned}
\epsilon_\theta(\tau_.^k, k, \mathcal{C}) &= f_L(f_{L-1}(...(f_1(\cdot)))) \\
O_{l+1} &= f_{l+1}(O_l, k; M_{i,l+1} \circ W_{l+1}), O_0 = \tau_{s_{z_q}}^{i,k}/\tau_{a_{z_q}}^{i,k},
\end{aligned}
\tag{5}
$$

where $\epsilon_\theta$ is the noise prediction model introduced in Equation 1, and $M_{i,l+1} \circ W_{l+1}$ represents the pairwise product. $\tau_{s_{z_q}}^{i,k}$ and $\tau_{a_{z_q}}^{i,k}$ denote the perturbed state or action sequences of task $i$ at diffusion step $k$. Through forward designing, we can selectively activate different weights for different tasks through the mask $M_{i,l+1}$, thus preserving previously acquired knowledge and reserving disengaged weights for other tasks. Though we can expediently calculate the masked output $O_{l+1}$ during forward propagation with weights or neurons masking, it poses a challenge to distinguishing the dependency from weights to loss and updating the corresponding weights during the backward propagation. In order to update the corresponding weights, we realize two methods. 1) Intuitively, we propose to update the neural network with the sparse optimizer rather than the dense optimizer [15], where the position and values of the parameters are recorded to update the corresponding weights. However, in the implementation, we find that the physical time consumption of the sparse optimizer is intractable (Refer to Table 9 of Appendix B.8 for more details.), which encourages us to find a more straightforward and convenient method. 2) Thus, we propose extracting and assembling the corresponding weights at the end of the training rather than updating the corresponding weights during training. This choice brings two benefits: (1) It can significantly reduce the time consumption spent on training. (2) It is friendly to implementation on complex network structures.

**Weights Assembling.** Assembling weights after training permits us to save the total acquired knowledge and do not need extra memory budgets. Concretely, after training on task $i$, we will obtain the weights $W_i$, which can be extracted with the mask $M_i$ from the total weights $W[i * \Omega]$, including all the diffusion model weights. We use $W_i$ to denote the weights related to task $i$, $\Omega$ is the training step on each CL task, and $W[i * \Omega]$ represents the total weight checkpoint at training step $i * \Omega$. Then, at the end of the training, we can assemble weights $\{W_i | i \in I\}$ by simply adding these weights together because of the exclusiveness property, i.e., $W = \sum_{i=1}^{I} W_i = \sum_{i=1}^{I} M_i \circ W[i * \Omega]$.

## 5 Experiments

In this section, we will introduce environmental settings, evaluation metrics, and baselines in the following sections. Then, we will report and analyze the comparison results, ablation study, and parameter sensitivity analysis. Other implementation details are shown in Appendix A.2 and A.3.

### 5.1 Environmental Settings

Following previous studies [98], we select MuJoCo Ant-dir and Continual World (CW) to formulate traditional CL settings with the same state and action spaces. In Ant-dir, we select 10-15-19-25 and 4-18-26-34-42-49, for training and evaluation. In CW, we adopt the task setting of CW10, which contains 10 robotic manipulation tasks. Additionally, we propose to leverage D4RL tasks [18] to construct the CL settings with diverse state and action spaces, where the task datasets in D4RL (Hopper, Walker2d, and HalfCheetah) contains 6 difficulty settings (random, medium, expert, medium-expert, medium-replay, and full-replay).

### 5.2 Evaluation Metrics

Considering the various reward structures of different environments, we should adopt different performance comparison metrics. For Ant-dir, we adopt the average episodic return over all tasks as the performance comparison, i.e., the final performance $P = \text{mean}(\sum_i R_i)$ is calculated based on the task $i$'s return $R_i$. In the CW environment, previous works [95, 6] usually adopt the success rate $\Psi$ as the performance metric. Thus, we adopt the average success rate on all tasks as the final performance, i.e., $P = \text{mean}(\sum_i \Psi_i)$. For the D4RL environments, we use the normalized score $\Phi$ [91, 42] as the metric to calculate the performance $P = \text{mean}(\sum_i \Phi_i)$, where $\Phi_i = \frac{R_i - R_{random}}{R_{expert} - R_{random}} * 100$. Usually, we can use the interface of these environments to obtain the score expediently.

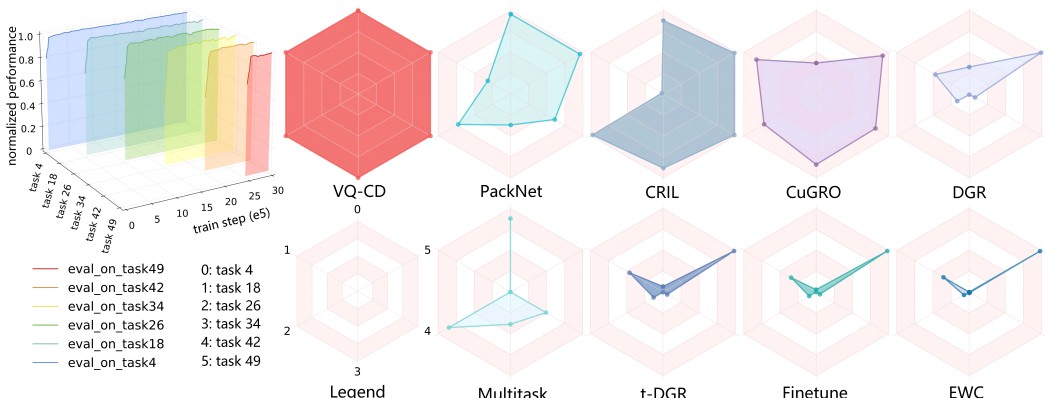

Figure 3: The comparison of VQ-CD and several baselines on the continual tasks setting (Ant-dir task 4-18-26-34-42-49). We train on each task for 500k steps. We report the normalized evaluation performance of VQ-CD in the top left corner, where the coordinates, e.g., task 4, represent evaluation on task 4 at different training tasks. To show the overall performance on all tasks, we show the normalized evaluation performance on the six tasks after finishing the training at the right part.

Table 2: The feature difference between the aligned features produced by the space alignment module. We randomly sample thousands of aligned state and action features to calculate the difference.

| Method | VQ-CD | | AE-CD | |
|---|---|---|---|---|
| feature difference | state difference | action difference | state difference | action difference |
| [Hopper-fr,Walker2d-fr,Halfcheetah-fr] | $8.83_{\pm1.98}$ | $4.54_{\pm0.74}$ | $51.31_{\pm26.91}$ | $14.06_{\pm2.09}$ |
| [Hopper-mr,Walker2d-mr,Halfcheetah-mr] | $9.03_{\pm1.97}$ | $4.45_{\pm0.74}$ | $48.12_{\pm21.94}$ | $15.39_{\pm3.71}$ |
| [Hopper-m,Walker2d-m,Halfcheetah-m] | $8.53_{\pm1.56}$ | $4.22_{\pm0.79}$ | $42.27_{\pm24.29}$ | $13.59_{\pm2.63}$ |
| [Hopper-me,Walker2d-me,Halfcheetah-me] | $8.93_{\pm2.00}$ | $4.05_{\pm0.56}$ | $57.91_{\pm36.94}$ | $13.93_{\pm3.20}$ |

## 5.3 Baselines

We select various representative CL baselines, which can be classified into diffusion-based and non-diffusion-based methods. For example, the diffusion-based methods consist of CRIL [21], DGR [80], t-DGR [99], MTDIFF [25], CuGRO [63], CoD [35], and CoD variants. The non-diffusion-based methods include L2M [79], EWC [55], PackNet [66], Finetune, IL-rehearsal [88], and Multitask. From the perspective of mainstream CL classification standards, these baselines can also be sorted as rehearsal-based methods (CRIL, DGR, t-DGR, CoD, and IL-rehearsal), regularization-based methods (L2M, EWC, CuGRO, and Finetune), and structure-based methods (PackNet, Multitask, and MTDIFF).

## 5.4 Experimental Results

In this section, we mainly separate the experimental settings into two categories, the traditional CL settings with the same state and action spaces and the arbitrary CL settings with different state and action spaces, to show the effectiveness of our method. Besides, we also investigate the influence of the alignment techniques, such as auto-encoder, variational auto-encoder, vector-quantized variational auto-encoder (we adopt this in our method). More deeply, we investigate how to deal with the potential demand for additional tasks beyond the pre-defined task length by releasing nonsignificant masks or expanding more available weights (Refer to Appendix B.7 for more details.).

The traditional CL settings correspond to the first question we want to answer: *Can VQ-CD achieve superior performance compared with previous methods in the traditional CL tasks?*

We use Ant-dir and Continual World [98] to formulate the continual task sequence, where we select two types of task sequence in Ant-dir and "hammer-v2, push-wall-v2, faucet-close-v2, push-back-v2, stick-pull-v2, handle-press-side-v2, push-v2, shelf-place-v2, window-close-v2, peg-unplug-side-v2" to construct CW10 CL setting. For simplicity, we do not align the state and action spaces with quantized alignment techniques because the traditional CL setting naturally has the same spaces.

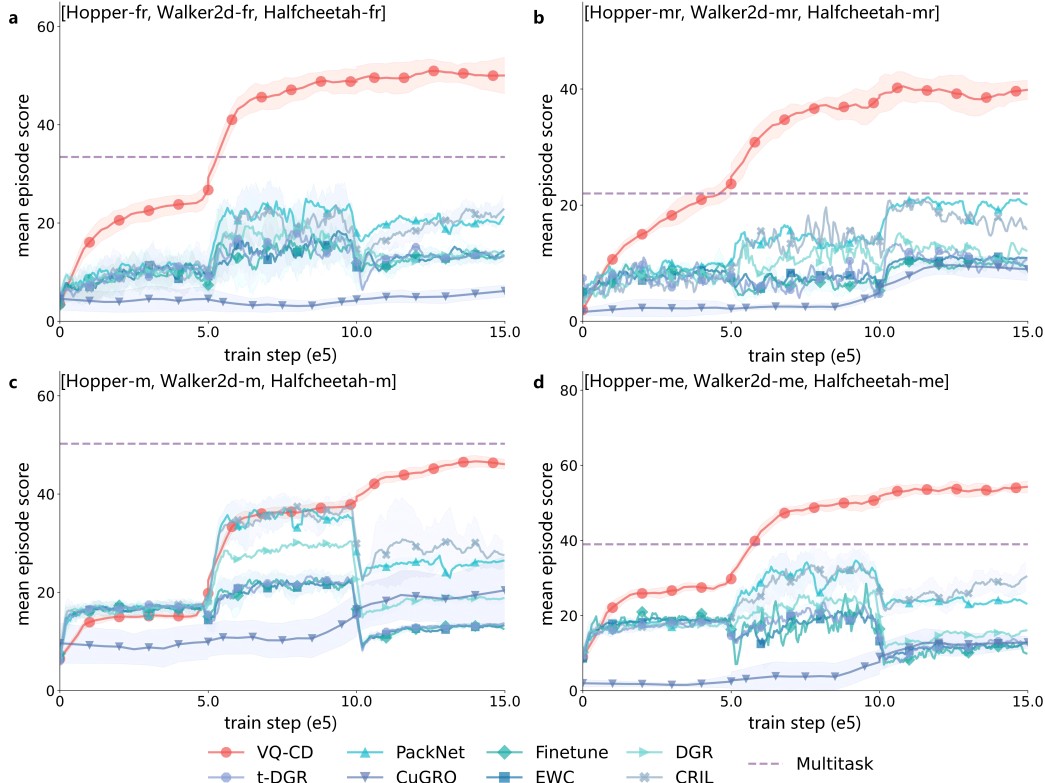

Figure 4: The comparison on the arbitrary CL settings. We select the D4RL tasks to formulate the CL task sequence. In order to align the state and action spaces, we use the pre-trained QSA module (the same as our method) to provide aligned spaces during training. The experiments are conducted on various dataset qualities, where the results show that our method surpasses the baselines not only at the expert datasets but also at the non-expert datasets, which illustrates the wide task applicability of our method. The datasets characteristic "fr", "mr", "m", and "me" represent "full-replay", "medium-replay", "medium", and "medium-expert", respectively. "Hopper", "Walker2d", and "Halfcheetah" are the different environments.

The comparison results between our method and several diffusion-based baselines are shown in Table 1, where these baselines include rehearsal-based (CoD and IL-rehearsal), parameter-sharing (CoD-LoRA), multitask training (Multitask CoD and MTDIFF), and representative diffusion RL methods (Diffuser-w/o rehearsal, CoD-RCR, and DD-w/o rehearsal). Our method surpasses all baselines in the Ant-dir setting by a large margin in Figure 3, which directly shows the effectiveness of our method. As another experiment of CL setting with the same state and action spaces, we report the results in Figure 11. Compared with the upper bound performance of Multitask, our method reaches the same performance after the CL training. With the increase in new tasks, our method continually masters new tasks and sustains the performance, while the baselines show varying degrees of performance attenuation, which can be found in the fluctuation of the curves. Moreover, the final performance difference between one method and the Multitask method indicates the forgetting character, which can be reflected by the overall upward trend of these curves. More experiments of shuffling task orders can be found in Appendix B.2.

The arbitrary CL settings correspond to the second question we want to answer: *Can we use the proposed space alignment method to enable VQ-CD to adapt to incoming tasks with various spaces?*

To answer the above question, we select D4RL to formulate the CL task sequence because of the various state and action spaces, and the results are shown in Figure 4. Considering the dataset qualities of D4RL [18], we choose different dataset quality settings and report the mean episode score that is calculated with $\frac{R_i - R_{random}}{R_{expert} - R_{random}} * 100$. Generally, from the four sub-experiments (**a**, **b**, **c**, and **d**), we can see that our method (VQ-CD) surpasses these baselines in all CL settings. Especially in the CL settings (Figure 4 **a** and **b**), where the datasets contain

low-quality trajectories, our method achieves a large performance margin even compared with the Multitask method. We can attribute the reason to the return-based action generation that helps our method distinguish different quality trajectories and generate high-reward actions during evaluation, as well as the selective weights activation that can reserve the previous knowledge and reduce forgetting. While other methods just possess the ability to continue learning and lack the ability to separate different qualities and actions, thus leading to poor performance. For trajectory qualities that are similar across the datasets (Figure 4 **c** and **d**), we can see lower improvement gains between our method and baselines. However, it should be noted that our method can still reach better performance than other baselines. Apart from the pre-trained QSA alignment, we also conduct experiments (Figure 12) on baselines that adopt padding to align state and action spaces in Appendix B.6.

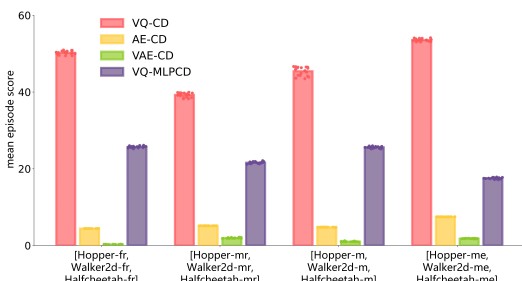

Figure 5: The ablation study of space alignment module and diffusion network structure. For each type of ablation study, we fix the other same and retrain the model on four D4RL CL settings.

## 5.5 Ablation Study

In this section, we want to investigate the influence of different modules of VQ-CD. Thus, the experiments contain two investigation directions: space alignment module ablation study and diffuser network structure ablation study. To show the importance of vector quantization, we change the space alignment module with auto-encoder (AE) and variational auto-encoder (VAE). Based on this modification, we retrain our method and report the results in Figure 5. The results show significant improvements in the D4RL CL settings, illustrating the importance and effectiveness of vector quantization in our method. Compared with AE-CD, VAE-CD performs poorly on all D4RL CL settings. The reason lies in that the implicit Gaussian constraint on each dimension may hurt the space alignment. Compared with the codebook in VQ-CD, AE-CD may cause a bigger difference between aligned features produced by AE (shown in Table 2), posing challenges for the diffusion model to model the distribution of the aligned features and leading to low performance. As for the diffuser network structure, we conduct the selective weights activation on the mlp-based and unet-based diffusion models. The latter structure is beneficial to making decisions with temporal information inside the trajectories, leading to higher performance evaluation.

## 5.6 Parameter Sensitivity Analysis

When performing on the aligned feature with diffusion models, the hyperparameters of the state and action of the quantized spaces alignment module matter. Usually, the complexity of states is more significant than the actions, so the codebook size controls

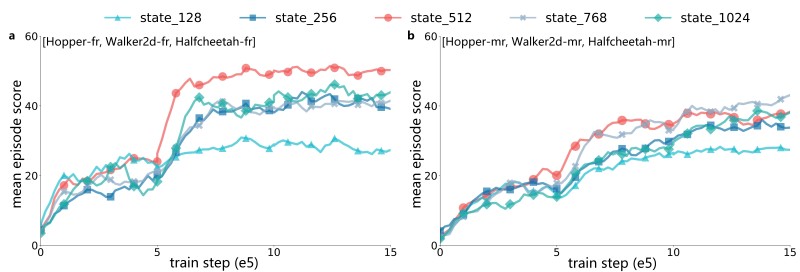

Figure 6: The effects of different codebook sizes about the states.

the performance of reconstruction. Thus, we investigate the effect of different codebook sizes and report the results in Figure 6. Obviously, a small codebook size limits performance, and a negative effect arises when it exceeds a certain value, such as 512. Additionally, we also conduct the experiments of parameter sensitivity analysis on actions latent and put the resutls in Figure 10, Appendix B.3.

# 6 Discussion

**The Interplay of VQ and CD.** In this paper, we investigate broadening the application scenarios of the same state and action spaces to tasks of arbitrary state and action spaces by space alignment. Vector quantization is verified as one effective way to achieve space alignment compared with AE, VAE, and padding. Furthermore, we adopt the diffusion model to perform continual learning based on VQ due to its strong model expressiveness and competitive performance. The ablation study illustrates that integrating VQ and CD induces the proposed powerful method VQ-CD.

**The Intuition of Constraint in QSA Module.** In Equation (4), We add a constraint to encourage a more concentrated distribution of the quantized representation vectors as shown in Table 2, which benefits the diffusion model in learning the data distribution in a limited range [30, 4]. However, this may not necessarily benefit other methods that do not focus on modeling distributions (Refer to Figure 4) because concentrated representations can make originally dissimilar state and action vectors from different tasks appear more similar, making them harder to distinguish and learn. We use the clip operation rather than convert the constraint to a penalty because our goal is to ensure that the magnitude of the quantized representation vectors does not exceed a certain value, rather than minimize the norm of the constraint. More discussion can be found in Appendix C.

# 7 Conclusion and Limitation

In this paper, we propose Vector-Quantized Continual Diffuser, called VQ-CD, which opens the door to training on any CL task sequences. The advantage of this general ability to adapt to any CL task sequences stems from the two sections of our framework: the selective weights activation diffuser (SWA) module and the quantized spaces alignment (QSA) module. SWA preserves the previous knowledge by separating task-related parameters with task-related masking. QSA aligns the different state and action spaces so that we can perform training in the same aligned space. Finally, we show the superiority of our method by conducting extensive experiments, including conventional CL task settings (identical state and action spaces) and general CL task settings (various state and action spaces). The results illustrate that our method achieves the SOTA performance by comparing with 17 baselines on 15 continual learning task settings. For limitations, our method belongs to task-aware CORL and is not suitable for task boundary-agnostic CORL, where an additional mechanism is needed to detect whether a task change has occurred. Clearly, this demands extra task similarity measurement mechanisms for detection. This is currently a limitation of our approach. However, we are confident that further progress will be reflected in our future research.

## Acknowledgement

We would like to thank Lijun Bian for her contributions to the figures and tables of this manuscript. We thank Siyuan Guo for his contributions to the writing suggestions of this manuscript. This work is supported in part by the National Key R&D Program of China (No. 2023YFF0905400, No. 2021ZD0112500); the National Natural Science Foundation of China (No. 62476110, No. U2341229); the National Key R&D Program of China (No. 2023YFF0905400, No. 2021ZD0112500); the Key R&D Project of Jilin Province (No. 20240304200SF); NSFC Grant (No. 62576364).

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

## A  Algorithm

### A.1  Pseudocode of VQ-CD

---

**Algorithm 1:** Vector-Quantized Continual Diffuser (VQ-CD)

---

**Input:** Noise prediction model $\epsilon_\theta$, inverse dynamic model $f_{inv,\psi}$, state and action quantized model $f_q(\theta_e, \theta_d, \theta_q)$, tasks set $\mathcal{M}_i, i \in \{1, ..., I\}$, each task training step $\Omega$, max diffusion step $K$, sequence length $T_e$, state dimension $d_s$, action dimension $d_a$, reply buffer $D_i, i \in \{1, ..., I\}$, noise schedule $\alpha_{0:K}$ and $\beta_{0:K}$

**Output:** $\epsilon_\theta, f_{inv,\psi}, \theta_e, \theta_d, \theta_q$

1 **Initialization:** $\theta, \psi, \theta_e, \theta_d$, and $\theta_q$

2 Separate the state-action trajectories of $D_i, i \in \{1, ..., I\}$ into state-action sequences with length $T_e$ and calculate the discounted returns $R_t^i = \sum_{t'=t}^{\infty} \gamma^{t'-t} r_{t'}$ for each step $t$

3 **for** *each task i* **do**

4     // **Quantized Spaces Alignment (QSA) Pretraining**

5     **for** *each train epoch* **do**

6         **for** *each train step* **do**

7             Sample states and actions from task $i$'s buffer $D_i$

8             Calculate the quantization loss and reconstruction loss

9             Updating the parameters $\theta_e$ of $f_{VQE}^i(\cdot; \theta_e)$, $\theta_d$ of $f_{VQD}^i(\cdot; \theta_d)$, and $\theta_q$ of $f_{\theta_q}^i(\cdot)$ by solving the problem of Equation 4

10         **end**

11     **end**

12     Save the task $i$'s well-trained $f_{VQE}^i(\cdot; \theta_e)$, $f_{VQD}^i(\cdot; \theta_d)$, and $f_{\theta_q}^i(\cdot)$

13     // **Selective Weights Activation (SWA) Diffuser Training**

14     Generate the task-related mask $M_i$ for task $i$

15     **for** *each train epoch* **do**

16         **for** *each train step m* **do**

17             Sample $b$ sequences $\tau_i^0 = \{s_{t:t+T_e}^i, a_{t:t+T_e}^i, R_{t:t+T_e}^i\} \in \mathbb{R}^{T_e \times (d_s+d_a+1)}$ from task $i$'s buffer $D_i$

18             Obtain the quantized state and action feature $s_{z_q}^i = f_{s,\theta_q}^i(f_{VQE_s}^i(s^i; \theta_e))$ and $a_{z_q}^i = f_{a,\theta_q}^i(f_{VQE_a}^i(a^i; \theta_e))$ with the QSA module

19             Train the inverse dynamic model $f_{inv}$ according to Equation 3

20             Formulate $s_{z_q}^i, a_{z_q}^i$ as sequences $\tau_{z_q}^{i,0} = \{s_{z_q,t:t+T_e}^i, a_{z_q,t:t+T_e}^i\}$

21             Sample the corresponding discounted returns $R_{t:t+T_e}^i$ from task $i$'s buffer $D_i$

22             Sample diffusion time step $k \sim \text{Uniform}(K)$ and return coefficient $b \sim \mathcal{B}(\lambda)$

23             Sample Gaussian noise $\epsilon \sim \mathcal{N}(0, \boldsymbol{I}), \epsilon \in \mathbb{R}^{b \times T_e \times d_{s_{z_q}}}$

24             Obtain $\tau_{s,z_q}^{i,k} = \sqrt{\bar{\alpha}_k} \tau_{s,z_q}^{i,0} + \sqrt{1-\bar{\alpha}_k}\epsilon$

25             Perform the forward propagation with Equation 5

26             Train $\epsilon_\theta$ according to Equation 1

27         **end**

28     **end**

29     Save task $i$'s related models as $\epsilon_{i*\Omega,\theta}$

30 **end**

31 // **Weights Assembling**

32 Construct new models $\tilde{\epsilon}_\theta$ with the same structure as $\epsilon_\theta$

33 **for** *each task i* **do**

34     Extract the task-related parameters $W_i$ with mask information $M_i$ from $\epsilon_{i*\Omega,\theta}$

35     Fill the corresponding task-related parameters $W_i = M_i \circ W_i$ into $\tilde{\epsilon}_\theta$

36 **end**

---

**Algorithm 2:** Evaluation Process

---

1  **for** *For each environmental step $t$ in task $i$* **do**
2     Receive the environmental state $s_t^i$
3     Set the return condition $R = 0.8$, $s_{t,z_q} = f_{s,\theta_q}^i(f_{VQE_s}^i(s_t^i; \theta_e))$
4     Construct $\hat{\tau}_{s_{z_q}}^K = [s_{t,z_q}, \hat{s}_{t+1,z_q}^K, \hat{s}_{t+2,z_q}^K, ...]$, where $\hat{s}_{t',z_q}^K \sim \mathcal{N}(0, \boldsymbol{I})$ for $t' > t$.
5     **for** *For $k$ from $K$ to $1$* **do**
6         Calculate $\hat{\epsilon}$ with $\epsilon_\theta$
7         Obtain $\hat{\tau}_{s_{z_q}}^{k-1}$ with Equation 2
8         Replace the first state of $\hat{\tau}_{s_{z_q}}^{k-1}$ with $s_{t,z_q}$
9     **end**
10    Extract $[s_{t,z_q}, \hat{s}_{t+1,z_q}]$ from $\hat{\tau}_{s_{z_q}}^0$
11    Obtain $a_{t,z_q} = f_{inv}(s_{t,z_q}, \hat{s}_{t+1,z_q})$
12    Interact with $a_t^i = f_{VQD_a}^i(a_{t,z_q}; \theta_d)$
13  **end**

---

The training of VQ-CD (Pseudocode is shown in Algorithm 1) contains three stages. 1) We first pre-train the QSA module for space alignment, as shown in lines 4-12, where we mainly want to solve the constrained problem of Equation 4. 2) Then, in lines 13-29, for each task $i$, we generate the task-related mask $M_i$ followed by a standard diffusion model training process (Refer to Equation 1 and Equation 5 for the training loss) on the aligned state and action spaces. 3) Finally, we assemble the task-related weights $W_i$ together with the mask information $\{M_i | i \in [1 : I]\}$ according to $W = \sum M_i \circ W[i * \Omega]$, where $\Omega$ is the training steps for each CL task, and $W[i * \Omega]$ is the weights checkpoints of $\epsilon_{i*\Omega,\theta}$. It is noted that the pre-training of the QSA module and the training of the SWA module can be merged together, i.e., for each task $i$, we can first train the QSA module related to task $i$ and then train the SWA module. The source code is available at https://github.com/JF-Hu/Vector_Quantized_Continual_Diffuser.

### A.2 Hyperparameters

We classify the hyperparameters shown in Table 3 into three categories: QSA module-related, SWA module-related, and training-related hyperparameters. We use the learning rate schedule when pre-training the QSA module, so the VQ learning rate decreases from 1e-3 to 1e-4. In our experiments, the maximum diffusion steps are set as 200, and the default structure is Unet. Usually, it is time-consuming for the diffusion-based model to generate actions in RL. Thus, we consider the speed-up technique DDIM [83] and realize it in our method to improve the generation efficiency during evaluation. For all models, we use the Adam [54] optimizer to perform parameter updating.

### A.3 Computation

We conduct the experiments on NVIDIA GeForce RTX 3090 GPUs and NVIDIA A10 GPUs, and the CPU type is Intel(R) Xeon(R) Gold 6230 CPU @ 2.10GHz. Each run of the experiments spanned about 24-72 hours, depending on the algorithm and the length of task sequences.

### A.4 Generation Speed-up Technique

The time and memory consumption of diffusion models is attributed to the mechanism of the diffusion generation process that requires multiple computation rounds to generate data [30]. Fortunately, previous studies provide useful speed-up strategies to accelerate the generation process [72, 83]. In this paper, we adopt DDIM as the default generation speed-up technique and reduce the reverse diffusion generation step to 10 compared to the original 200 generation steps. In Table 4, we use the CL setting of Ant-dir task-4-18-26-34-42-49 as an example to compare the time consumption of different generation steps. Compared with the original 200 diffusion steps, we can see that incorporating DDIM will significantly (**19.76×**) improve the efficiency of generation. In the experiments, we find that 10 diffusion steps setting performs well on performance and generation efficiency. Thus, we set the default sampling speed-up stride to 20, and the diffusion step is 200/20=10 steps.

Table 3: The hyperparameters of VQ-CD.

| | Hyperparameter | Value |
|---|---|---|
| QSA section | network backbone | MLP |
| | hidden dimension of QSA module | 256 |
| | commitment cost coefficient | 0.25 |
| | codebook embedding limit $\rho$ | 3.0 |
| | state codebook size per task | 512 |
| | number of state latent | 10 |
| | state latent dimension | 2 |
| | action codebook size per task | 512 |
| | number of action latent | 5 |
| | action latent dimension | 2 |
| | alignment type | VQ/AE/VAE |
| | VQ learning rate | [1e-4,1e-3] |
| SWA section | network backbone | Unet/MLP |
| | hidden dimension | 256 |
| | sequence length $T_e$ | 8 |
| | diffusion learning rate | 3e-4 |
| | guidance value | 0.95 |
| | mask rate | $1/I$ |
| | condition dropout $\lambda$ | 0.25 |
| | max diffusion step $K$ | 200 |
| | sampling speed-up stride | 20 |
| | condition guidance $\omega$ | 1.2 |
| | sampling type of diffusion | DDIM |
| Training | loss function | MSE |
| | batch size | 32 |
| | optimizer | Adam |
| | discount factor $\gamma$ | 0.99 |

Table 4: The comparison of generation speed with different generation steps under the CL setting of Ant-dir task-4-18-26-34-42-49. In the main body of our manuscript, we use the 10 diffusion steps setting for all experiments.

| Diffusion steps | 200 (original) | 100 | 50 | 25 | 20 | 10 |
|---|---|---|---|---|---|---|
| sampling speed-up stride | 1 (original) | 2 | 4 | 8 | 10 | 20 |
| Time consumption of per generation (s) | $5.73_{\pm0.29}$ | $2.88_{\pm0.21}$ | $1.41_{\pm0.16}$ | $0.71_{\pm0.18}$ | $0.58_{\pm0.17}$ | $0.29_{\pm0.15}$ |
| Speed-up ratio | 1× | 1.99× | 4.06× | 8.07× | 9.88× | 19.76× |

## A.5 Computational Cost Analysis

In Table 5, we report the GPU memory consumption during the training process. We mainly consider the experiments on the D4RL, Ant-dir, and CW CL tasks. We can change the first block of the diffusion model to make our model suitable for a longer CL task sequence. For example, we expand the dimension length from 512 to 1024 when switching the CL training task from 'task-10-15-19-25' to 'task-4-18-26-34-42-49'.

We compare the computational cost, including generation time and memory consumption, with diffusion-based methods, such as CuGRO, and transformer-based methods, such as L2M, and report the results in Table 6 and Table 7. The results show that, compared to the baselines, our method achieves lower time overhead and better performance with similar memory usage.

Table 5: The GPU memory consumption.

| domain | CL task setting | GPU memory consumption (GB) |
|---|---|---|
| D4RL | [Hopper-fr,Walker2d-fr,Halfcheetah-fr]
[Hopper-mr,Walker2d-mr,Halfcheetah-mr]
[Hopper-m,Walker2d-m,Halfcheetah-m]
[Hopper-me,Walker2d-me,Halfcheetah-me] | 4.583
4.583
4.583
4.583 |
| Ant-dir | task-10-15-19-25
task-4-18-26-34-42-49 | 4.711
5.955 |
| CW | CW10 | 5.897 |

Table 6: The computational cost of generation speed with different generation steps in D4RL [Hopper-m,Walker2d-m,Halfcheetah-m] tasks.

| method | base | VQ-CD | CuGRO |
|---|---|---|---|
| time consumption | 5.73 | 0.29 | 0.33 |
| speed-up ratio | 1× | 19.8× | 17.4× |
| score | - | 45.4 | 27.6 |

Table 7: The comparison of GPU memory consumption. We conduct the experiment with NVIDIA GeForce RTX 3090 GPUs and Intel(R) Xeon(R) Gold 6230 CPU @ 2.10GHz.

| domain | CL task setting | Method | GPU overhead (GB) | Parameters size (M) | Physical training time (h) | Performance |
|---|---|---|---|---|---|---|
| D4RL | [Hopper-fr,Walker2d-fr,Halfcheetah-fr] | VQ-CD | 4.6 | 89.1 | 55 | 48.0 |
| D4RL | [Hopper-fr,Walker2d-fr,Halfcheetah-fr] | L2M | 6.7 | 57.8 | 92 | 13.4 |
| D4RL | [Hopper-fr,Walker2d-fr,Halfcheetah-fr] | L2M-large | 8.8 | 96.0 | 94 | 16.0 |

## A.6 Baselines Implementation

All the comparison methods used in this paper utilize their official codebases. Specifically,

- For L2M, we use the official source code: https://github.com/ml-jku/L2M

- For CuGRO, we use the official source code: https://github.com/NJU-RL/CuGRO

- For CoD, we use the official source code: https://github.com/JF-Hu/Continual_Diffuser

- For MTDIFF, we use the official source code: https://openreview.net/forum?id=fAdMly4ki5

## A.7 Network Details

In the diffusion model (SWA module), we utilize a UNet network structure, incorporating residual connections at both the input and output of each block. Additionally, residual connections are applied between the down-sampling and up-sampling blocks, meaning that the output of the down-sampling block serves as the input to the up-sampling block. The convolution kernels in the UNet are one-dimensional, with their shapes corresponding to the shape of the mask matrix.

In the QSA module, there are no shared parameters. The primary purpose of the QSA module is to align the state and action spaces across different environments. Consequently, for different tasks, the internal components of the QSA module, vector quantized encoder (VQE), vector quantized decoder (VQD), and codebook are task-specific, and none of their parameters are shared. Thanks to the alignment provided by the QSA module, the inverse dynamics model in the SWA module can be shared. This is because the state and action spaces of different environments are mapped into an alignment space with the same value range.

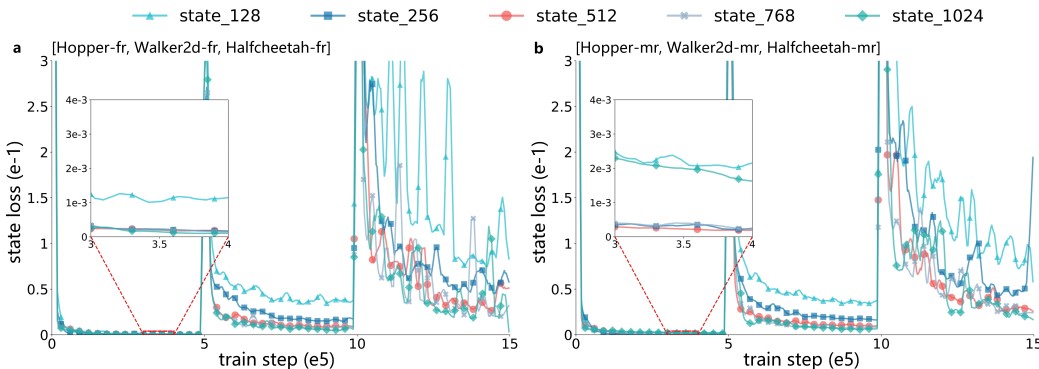

Figure 7: The QSA module loss under different codebook sizes about states. We explore five codebook size settings: 128, 256, 512, 768, and 1024. The red line represents the experimental codebook size setting for states.

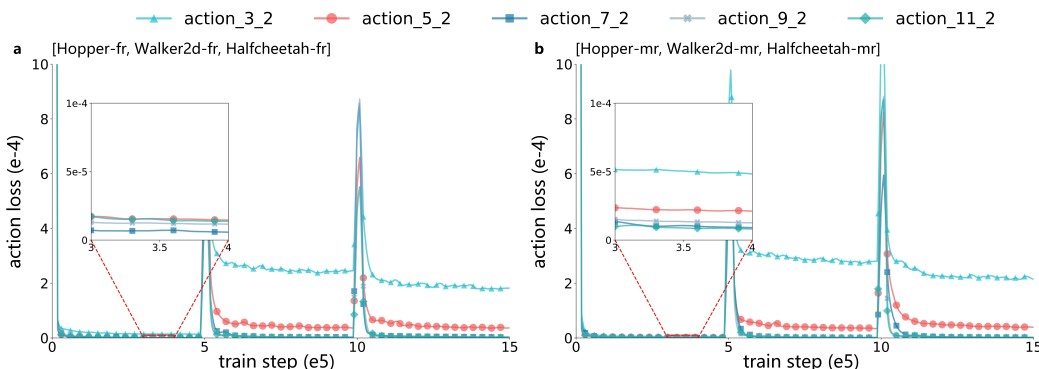

Figure 8: The QSA module loss under different latent numbers about actions. The setting includes 3, 5, 7, 9, and 11, which correspond to the aligned action space sizes 6, 10, 14, 18, and 22. The red line represents the experimental latent numbers setting for actions.

# B  Additional Experiments

## B.1  QSA Module Loss Analysis

Under the same hyperparameter settings in Section 5.6, we report the loss of the QSA Module to investigate the effects of codebook size and latent number. For the states, we investigate the influence of codebook size, where we set codebook size as 128, 256, 512, 768, 1024, and select D4RL CL setting [Hopper-fr, Walker2d-fr, Halfcheetah-fr] and [Hopper-mr, Walker2d-mr, Halfcheetah-mr] as the example. The results are shown in Figure 7, where we train the QSA module on each task for 5e5 steps. We can see that for states, a codebook size of 512 is good enough for aligning the different tasks' state spaces. A larger codebook size, such as 768 and 1024 in Figure 7 **a** and **b**, will not bring significant loss improvements. Smaller codebook sizes can not provide sufficient latent vectors to map the state spaces to a uniform space.

For the action, we select the latent number to explore the QSA action loss and report the results in Figure 8. We can see the same trend that has been seen in QSA state loss (Figure 7). Though the lower loss value of the more latent number indicates that we should use more action latent vectors, we find that the gap between action latent number settings 5 and 7 is small when we increase computation resources. Besides, we also see inconspicuous performance gains in the final performance in Figure 10, which urges us to use 5 as the default action latent number setting. For the action latent vector dimension, we directly use 2 as the default setting.

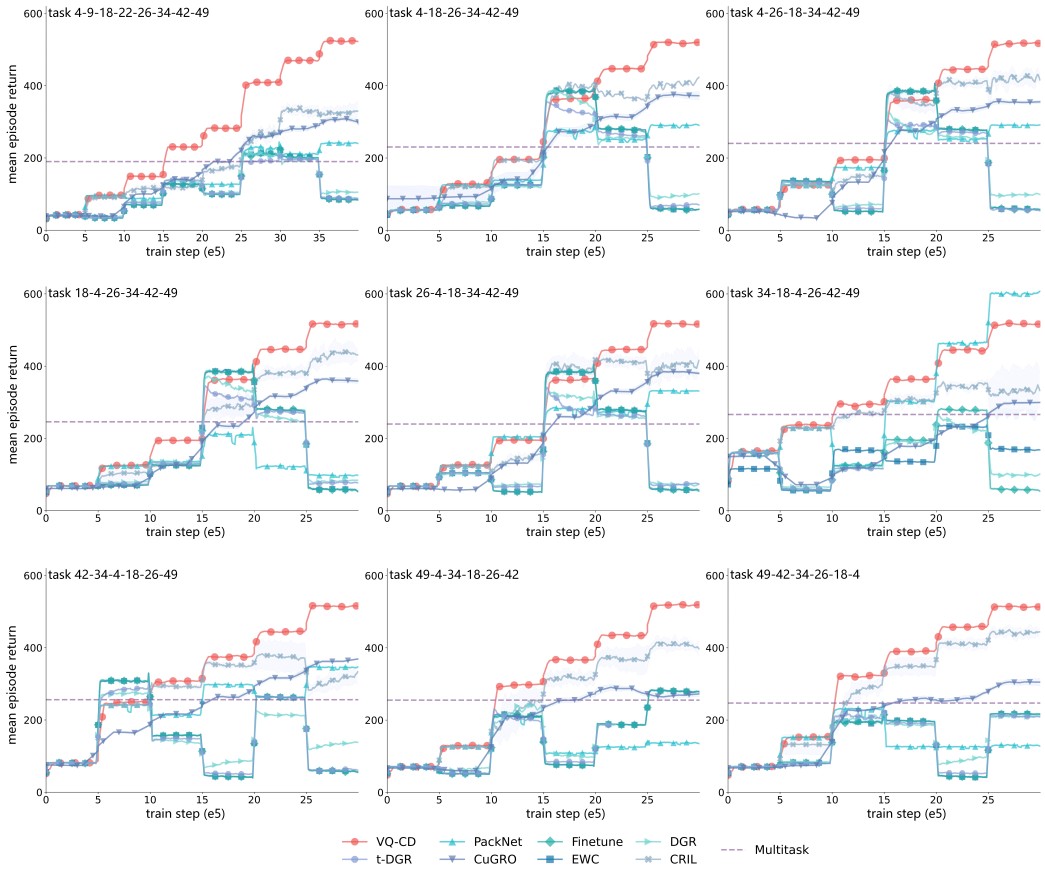

Figure 9: The experiments of Ant-dir with shuffled task order. We investigate the influence of shuffled task order in the Ant-dir environment, where the experiments include inserting new tasks into the predefined task order '4-18-26-34-42-49' and disrupting the tasks order.

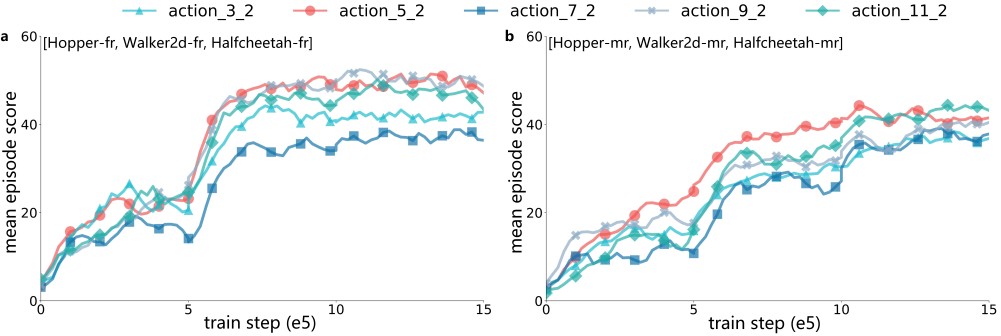

Figure 10: The effects of the number of latent vectors about the actions.

## B.2   Experiments of Task Order Shuffling

To investigate the influence of task order in CORL, we choose Ant-dir as the testbed and change the task order for new CL training. We change the task order by inserting new tasks into the predefined task order '4-18-26-34-42-49' and disrupting the task order. We can see from the results shown in Figure 9 that our method achieves the best performance in almost all CL task orders. The task order will affect the final performance of other baselines. For example, CRIL performs better in the task orders 'task 18-4-26-34-42-49' and 'task 49-42-34-26-18-4' than in other task order experiments. Another example is PackNet, which achieves the best performance only in the task order 'task

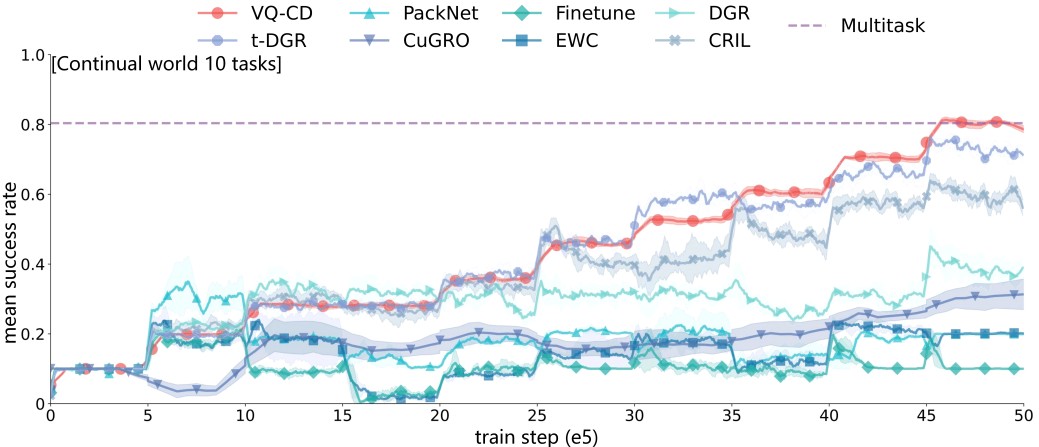

Figure 11: The experiments on the CW10 tasks, which contain various robotics control tasks. We train each method on each task for 5e5 steps and use the mean success rate on all tasks as the performance metric. Generally, we can see the superiority of our method from the above figure.

34-18-4-26-42-49'. Different from the baselines, whose performance fluctuates with the changing of task orders, our method (VQ-CD) shows stable training performance no matter what task orders are defined.

## B.3 Experiments of Parameter Sensitivity

In Figure 6 we investigate the effect of different codebook sizes and find that a small codebook size limits performance, and a negative effect arises when it exceeds a certain value. For the actions, we believe the actions can be decomposed into several small latent vectors, and the number of latent vectors is crucial for reconstructing actions. Similarly, we also see the same trend in Figure 10, which shows that more latent vectors are not always better.

## B.4 The Benefits of Inverse Dynamics

Following previous studies [4], the inverse dynamics is introduced to produce actions based on the state sequence generated by the diffusion model. We choose to model the distribution of the state sequence rather than the state-action sequence on the basis of two reasons: 1) Usually, in many robotics control scenarios, the actions are often represented as joint torques, which are high-frequency and less smooth, making it hard to model and predict the action sequence. 2) The state is usually continuous in RL, but the mode of action is diverse, such as discrete and continuous. Modeling state sequences separately makes the diffusion-based model more generic to extensive RL scenarios. Using the diffusion model to model the state sequences and producing actions with the inverse dynamics are not related to accommodating different action spaces across tasks.

To further investigate the benefit of producing actions with inverse dynamics rather than generating $(s, a)$ together with diffusion models, we conduct the experiments of modeling state and action sequences together with diffusion models and only modeling state sequences with diffusion models. Table 8 shows that when using inverse dynamics, our method can achieve better performance compared with directly producing action with diffusion models.

## B.5 Experiments on Continual World

We select CW10 as another experiment of CL setting with the same state and action spaces, where the task number is 10. We report the results in Figure 11. Compared with the upper bound performance of Multitask, our method reaches the same performance after the CL training. With the increase in new tasks, our method continually masters new tasks and sustains the performance, while the baselines show varying degrees of performance attenuation, which can be found in the fluctuation of the curves.

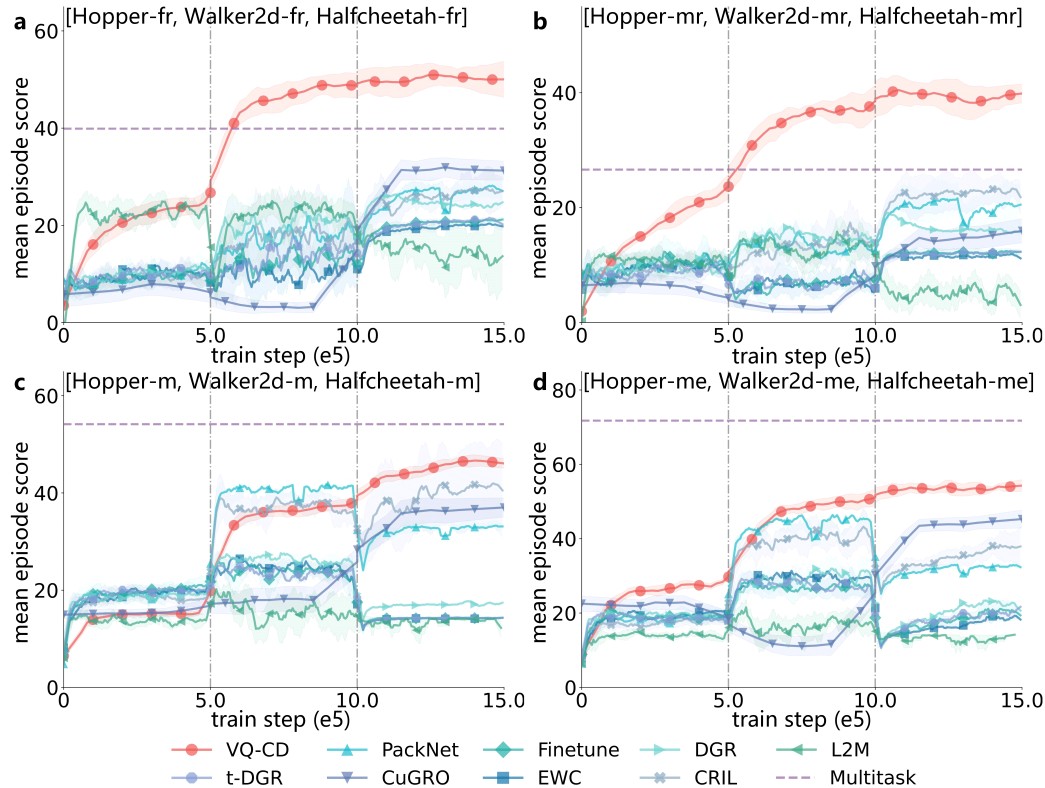

Figure 12: The comparison on the arbitrary CL settings. We select the D4RL tasks to formulate the CL task sequence. We leverage state and action padding to align the spaces. The experiments are conducted on various dataset qualities, where the results show that our method surpasses the baselines not only at the expert datasets but also at the non-expert datasets. The datasets characteristic "fr", "mr", "m", and "me" represent "full-replay", "medium-replay", "medium", and "medium-expert", respectively. "Hopper", "Walker2d", and "Halfcheetah" are the different environments.

Table 8: The comparison of producing actions with the diffusion model and the inverse dynamics.

| task | producing action with diffusion model | producing action with inverse dynamics |
|------|:---:|:---:|
| Ant-dir 4-18-26-34-42-49 | 498.2 | 524.1 |
| [Hopper-m,Walker2d-m,Halfcheetah-m] | 39.5 | 45.4 |

## B.6    Experiments of Baselines Equipped QSA

In Section 5.4, we report the comparison of our method and baselines in the arbitrary CL settings, where in the D4RL CL settings, we adopt the pre-trained QSA module to align the state and action spaces. Apart from the pre-trained QSA module, we can also use the state and action padding to align the different state and action spaces. In Figure 12, we report the results of baselines equipped with state and action padding. From the results, we can also see that our method still achieves the best performance compared with these baselines. Considering the results of Figure 12, Figure 5 (VQ-MLPCD), and Figure 4 (VQ baselines), we can see the importance of complementary sections: QSA and SWA.

## B.7    Supporting Tasks Training Beyond the Pre-defined Task Sequence

After training on pre-defined task sequences, we may hope the model has the capacity to support training on potential tasks, which means that we need more weights or weight masks. Releasing weight masks that are used to learn previous tasks is a straightforward choice when the total weights are fixed. We conduct the experiments of mask pruning on Ant-dir 'task 4-18-26-34-42-49' and

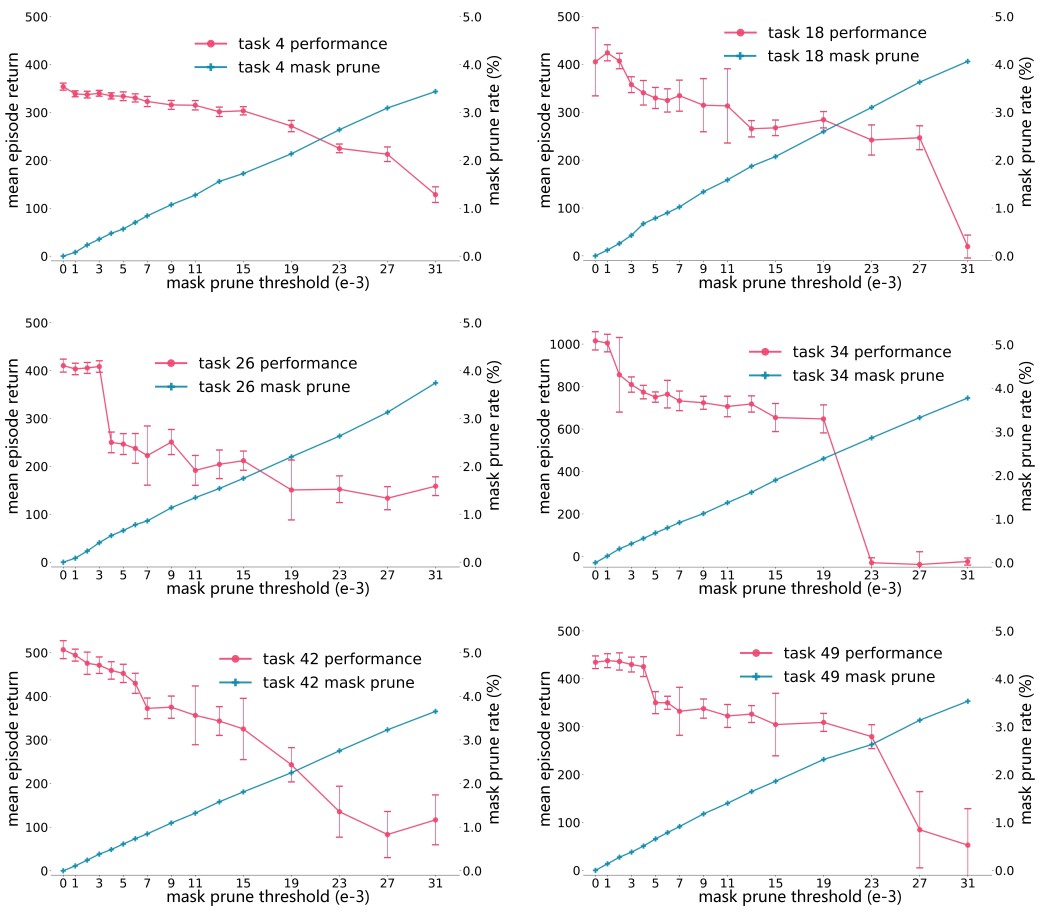

Figure 13: The mask pruning experiments of Ant-dir 'task 4-18-26-34-42-49'. We investigate the task pruning according to the absolute weight values, i.e., we release the weights to train on potential new tasks according to the mask prune threshold.

Table 9: The comparison of time consumption per update between sparse and dense (normal) optimizers. We compare these two types of optimizers on the CL settings and find that when we first use the normal optimizer, such as Adam, to train the model and then use weights assembling to obtain the final model, the total physical time consumption is significantly smaller than sparse optimizer (e.g., sparse Adam).

| domain | CL task setting | time consumption per update (s) | |
| --- | --- | --- | --- |
| | | dense optimizer | sparse optimizer |
| D4RL | [Hopper-fr,Walker2d-fr,Halfcheetah-fr] | $0.089_{\pm 0.219}$ | $0.198_{\pm 0.224}$ |
| | [Hopper-mr,Walker2d-mr,Halfcheetah-mr] | $0.096_{\pm 0.223}$ | $0.197_{\pm 0.223}$ |
| | [Hopper-m,Walker2d-m,Halfcheetah-m] | $0.089_{\pm 0.211}$ | $0.195_{\pm 0.224}$ |
| | [Hopper-me,Walker2d-me,Halfcheetah-me] | $0.090_{\pm 0.223}$ | $0.206_{\pm 0.225}$ |
| Ant-dir | task-10-15-19-25 | $0.062_{\pm 0.064}$ | $0.239_{\pm 0.282}$ |
| | task-4-18-26-34-42-49 | $0.064_{\pm 0.061}$ | $0.214_{\pm 0.270}$ |
| CW | CW10 | $0.061_{\pm 0.065}$ | $0.218_{\pm 0.286}$ |

report the performance and weight mask prune rate when pruning weight masks according to certain absolute value thresholds in Figure 13. The results illustrate that we can indeed release some weight masks under the constraint of preserving 90% or more performance compared with the unpruned model. On the other hand, we can also see that this mask pruning method can only provide finite

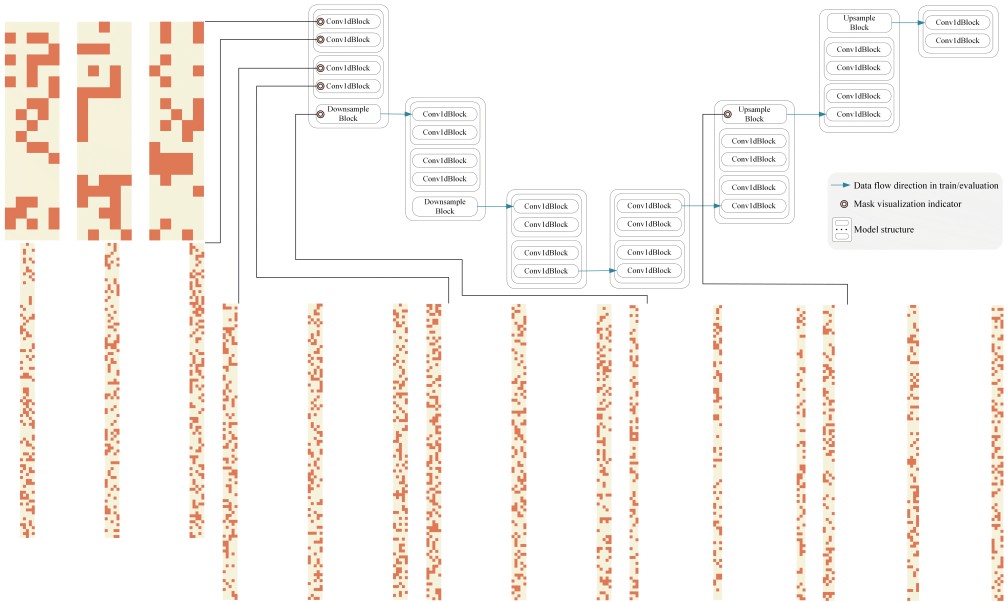

Figure 14: Mask matrices visualization. we select [Hopper-m, Walker2d-m, Halfcheetah-m] as an example to report the mask results. For each mask matrix, we only draw the first 100 channels of the weights mask matrix if the mask matrix is too large.

capacity for tasks beyond the predefined task sequence. We postpone the systematic investigation of mask pruning to future work.

### B.8 Time Consumption of Different Optimizers

In the CL settings of our experiments, we compare two types of optimizers and find that when we first use the normal optimizer, such as Adam, to train the model and then use weights assembling to obtain the final model, the total physical time consumption is significantly smaller than sparse optimizer (e.g., sparse Adam). Thus, we propose the weights assembling to obtain the final well-trained model after the training rather than suffering huge time burden of sparse optimizer during the training.

### B.9 Mask Visualization

We select [Hopper-m, Walker2d-m, Halfcheetah-m] to visualize the weights mask of our method in Figure 14. To make it easy to show the mapping relation between masks and the weights, we draw the network structure and mask matrices, where we only report the first 100 channels of the mask matrices.

### B.10 Alignment Space Visualization

In order to further demonstrate the effectiveness of our method. We conduct the visualization experiments of aligned state feature and report the visualization results in Figure 15. From the experimental results, we can see that the state features learned by the AE method are not well-mapped to separate regions but are instead mapped to multiple areas. In contrast, the features obtained by our method are better partitioned into individual regions, which is more conducive for the model to capture the data distribution.

## C   Further Discussion of Experiments

**The Interplay of VQ and CD.**   In this paper, we investigate broadening the application scenarios of the same state and action spaces to tasks of arbitrary state and action spaces by space alignment. Vector quantization is verified as one effective way to achieve space alignment compared with AE,

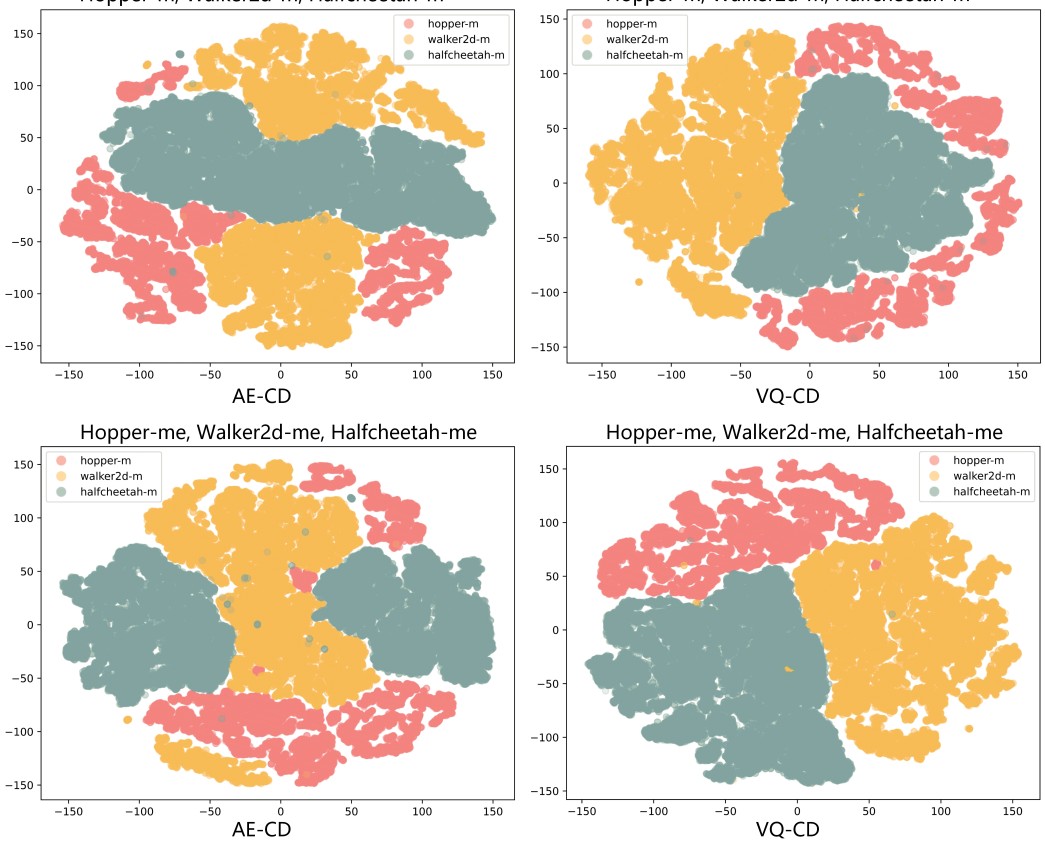

Figure 15: Visualization of aligned state feature. We use the QSA module to process the different state spaces and align them in the same space. Then we use t-SNE [87] to visualize aligned state features.

VAE, and padding. Furthermore, we adopt the diffusion model to perform continual learning based on VQ due to its strong model expressiveness and competitive performance. The ablation study illustrates that integrating VQ and CD induces the proposed powerful method VQ-CD.

**The Intuition of Constraint in QSA Module.** In Equation (4), We add a constraint to encourage a more concentrated distribution of the quantized representation vectors as shown in Table 2, which benefits the diffusion model in learning the data distribution in a limited range [30, 4]. However, this may not necessarily benefit other methods that do not focus on modeling distributions (Refer to Figure 4) because concentrated representations can make originally dissimilar state and action vectors from different tasks appear more similar, making them harder to distinguish and learn. We use the clip operation rather than convert the constraint to a penalty because our goal is to ensure that the magnitude of the quantized representation vectors does not exceed a certain value, rather than minimize the norm of the constraint.

**Further Discussion of Experiments.** In Figure 4 (a) and (b), we can see that VQ-CD surpasses Multitask. The reason is as follows. 1) The Action Quality Discrimination Ability Differences: The datasets contain trajectories collected from the entire training process, i.e., from a random policy to a well-trained policy. Our method leverages accumulated discounted returns to guide the generation of state sequences, encouraging the generation of higher-return state sequences. Consequently, the actions generated by the inverse dynamics model also yield higher returns. In contrast, Multitask does not currently incorporate returns, resulting in lower performance. In Figures 4 (c) and (d), the variance of trajectory returns in the dataset is smaller, allowing Multitask to achieve better learning outcomes. 2) Architecture Differences: The multitask baseline is implemented using the MLP architecture, which is often insufficient to model the complex state-action mappings across diverse tasks. In contrast, our VQ-CD method leverages a diffusion-based policy model, which has demonstrated superior expressiveness and stability in high-dimensional generative modeling. This difference leads

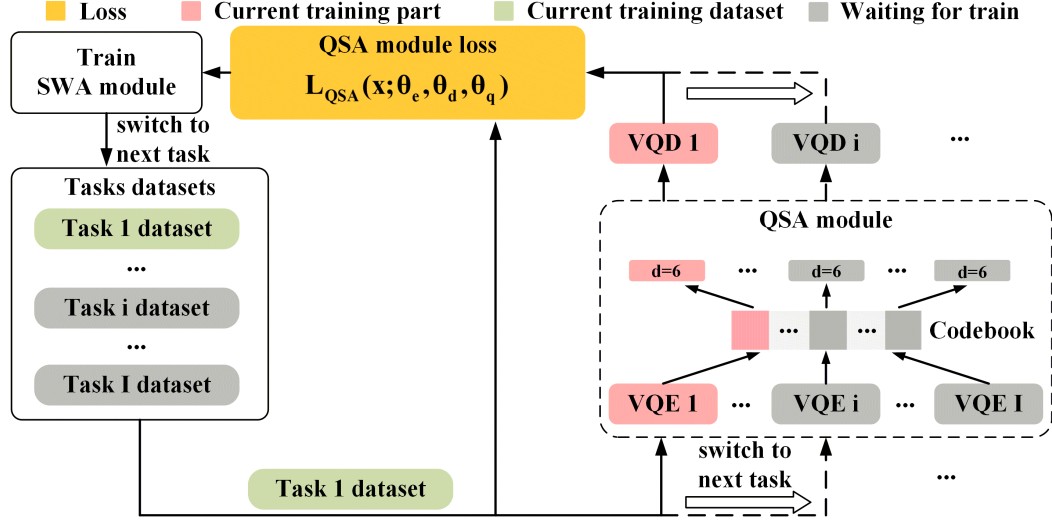

Figure 16: A graphical depiction of QSA training. From the figure, it is intuitively clear that the training of the QSA module can fully adhere to the CL training setup.

to the performance differences in the experiments. 3) Representation Ambiguity Across Tasks: As shown in Table 3, the QSA module maps task-specific state-action pairs into a shared aligned latent space, where the distances between different tasks' state- and action-aligned vectors are relatively small. This results in high semantic proximity across tasks. For Multitask training, all tasks' state- and action-aligned vectors are blended together, which makes it challenging to learn the mapping of a state-aligned vector to the correct action-aligned vector, leading to degraded performance. In contrast, VQ-CD separates alignment with QSA and generation via the conditional diffusion model, which enables more precise modeling and avoids such interference.

**The Motivation of Adopting Diffusion Models.** We select diffusion models as our foundational method primarily based on the following reasons:

- Natural support for multi-modal action distribution modeling: Diffusion models can effectively avoid the limitations of traditional Gaussian models that cannot model multi-peak action distributions.
- Powerful model expressiveness: Diffusion models support the modeling of complex data or trajectory distributions.
- Stable log-likelihood training: Diffusion models can effectively prevent mode collapse and training instability issues.
- Competitive model performance: Diffusion-based RL methods have also shown huge potential in many robotic control scenarios.

SWA is specifically designed for diffusion models with one-dimensional convolutional structures because diffusion models exhibit a different dependency pattern: every output of a given channel is influenced by all weights of the convolution kernel in that channel due to the weight-sharing nature of convolutions. Thus, traditional output neuron masking is not suitable for diffusion models with one-dimensional convolutional structures because we can not enable task-related parameters through masking certain output neurons. However, SWA applies masking directly on convolution kernel parameters, allowing us to selectively activate neurons in the convolution kernel to control the training of task-related parameters. QSA is applicable to any model. Different state and action spaces lead to significant distribution differences between tasks, making it impossible to use a single diffusion model for learning. QSA aligns these varying state and action spaces into the same latent space, enabling effective continual training under the same diffusion backbone.

## D   Discussion of Future Research Directions

**Adaptation to Dramatic Shifts across Tasks.** Our work currently focuses on tasks with certain similarities, rather than dramatic shifts in state representation formats across tasks. Usually, the

applicability of continual learning across tasks with completely different state spaces is quite narrow, and previous studies rarely explore this area [85]. The purpose of continual learning is to progressively master new tasks by discovering common knowledge between tasks. There are several substantial challenges facing high-dimensional image-based observations, particularly in terms of representation alignment, codebook generalization, and action generation. 1) For representation, we can introduce a visual encoder, e.g., ViT, to extract compact latent representations from images before passing them into the diffusion models, thereby aligning the modalities. 2) For the codebook, the QSA module can be modified to support modality-specific encoders while maintaining a shared codebook space or adopting modality-conditional quantization. 3) For the inverse dynamics, if we use inverse dynamics to directly produce actions with image-like states, then we need to upgrade inverse dynamics to a vision-conditioned architecture.

**Theoretical Analysis about the QSA and SWA.** We can see that the current version lacks theoretical proof to demonstrate that space-aligned representations can effectively extract common knowledge between tasks. However, our method has a theoretical foundation for the representation with vector quantization [77, 68]. In our paper, the QSA module maps tasks' states and actions into a shared discrete latent space using the codebook. This process enables structural alignment between otherwise heterogeneous observation spaces, making it easier for the diffusion model to generalize across tasks. We can obtain that under Robbins–Monro stepsizes and Lipschitz gradients, the expected gradient of the QSA loss $\lim_{n\to\infty} \mathbb{E}[||\nabla_{\theta_e,\theta_d}\mathcal{L}_{QSA}||]$ vanishes as the training iteration increases, where $n$ is the training iteration [86].

For the convergence proof of SWA, according to the previous studies of diffusion model [30], the denoising score-matching loss ($\mathcal{L}(\theta) = \mathbb{E}[||\epsilon - \epsilon_\theta(\tau_s^k, k, b * \mathcal{C})||_2^2]$) is a lower bound on the negative log-likelihood. Under the Lipschitz constraint, the loss landscape is smooth, and optimization using Adam or SGD with diminishing learning rates satisfies the Robbins–Monro conditions. Thus, the training process converges to a stationary point of the objective.

**Extension to General Online RL Tasks.** Our method is currently designed for the offline RL setting primarily due to the multiple-step sampling process inherent in diffusion models, which leads to significant computational latency during interaction in online RL. To enable online RL extensions in the future, several promising research lines can be explored: 1) Parallelized sampling techniques to reduce per-decision latency. 2) Advanced samplers (e.g., DDIM and DPM-solver) to accelerate generation. 3) Distilling the diffusion model into the consistency model that can realize single-step generation.

