# OpenReview forum: "Tackling Continual Offline RL through Selective Weights Activation on Aligned Spaces"
_NeurIPS.cc/2025/Conference — NeurIPS 2025 poster_

### Official Review · Reviewer_Yyw3 · 2025-07-01

**Clarity:** 2
**Significance:** 3
**Originality:** 3
**Rating:** 4
**Confidence:** 2

**Summary:**

This paper proposes VQ-CD, a framework for continual offline reinforcement learning that handles heterogeneous task spaces by combining vector quantization-based space alignment and selective weight activation. The method achieves strong performance across both traditional and arbitrary CL settings.

**Questions:**

Please see weaknesses 1- 5.

I am currently leaning toward a borderline reject. But I am open to increasing my score if the authors can adequately address these issues during the rebuttal.

**Ethical Concerns:**

["NO or VERY MINOR ethics concerns only"]

**Final Justification:**

Thank you for the rebuttal. It has addressed my concerns, and I will raise my score accordingly.

**Limitations:**

Yes

**Quality:**

3

**Strengths And Weaknesses:**

**Strength**
- The paper conducts comprehensive experiments across both traditional and arbitrary CL settings to demonstrate the effectiveness of the proposed VQ-CD framework.
- Detailed ablation studies provide evidence for the contribution of each module.

**Weaknesses**
- Some claims may be overstated. For example, Line 138 states that the method supports “our method enables training on **any** CL task sequence” and “QSA enables our method to adapt for **any** continual learning task setting.” However, it appears that VQ-CD assumes that task identities and boundaries are known. Can the method be applied in task-free CL settings?
- The SWA module contributes significantly even when task spaces are consistent. Could the authors elaborate on how SWA compares structurally and functionally to existing parameter-isolation methods such as PackNet?  Is there a risk of capacity saturation or degradation in performance as tasks accumulate, especially in longer task sequences?
-  The authors mention that each task occupies a separate region in the quantized space. As the number of tasks increases, how does this design affect the size and efficiency of the codebook? Is there any mechanism to reuse or compress unused entries, or does the method assume a large fixed codebook with sufficient capacity?
- In Figure 2, the performance of the multitask baseline, which is often considered the upper bound in continual learning, is unexpectedly poor. In Figure 3, VQ-CD even substantially outperforms multitask training. Could the authors provide insights or possible reasons for this?
- Given that VQ-CD builds a shared latent space across tasks, is the model able to generalize to unseen tasks with similar structure but new state/action configurations?

---

> ### Author Rebuttal · Authors · 2025-07-31
>
> Dear Reviewer Yyw3,
>
> We appreciate your further suggestions and respond to your concerns below.
>
> ### [I]. Response to Weaknesses
>
> > **[1/5] W1:** Some claims may be overstated. For example, Line 138 states that the method supports “our method enables training on any CL task sequence” and “QSA enables our method to adapt for any continual learning task setting.” However, it appears that VQ-CD assumes that task identities and boundaries are known. Can the method be applied in task-free CL settings?
>
> Thanks for your suggestions. We will revise the corresponding content to restrict the scope of application in task-aware continual learning. Detecting the task boundaries demands extra task similarity measurement mechanisms for detection. Admittedly, this is currently a limitation of our approach. It is one of the challenges we are actively addressing.
>
> Task-aware continual learning is quite prevalent in many applications, such as CW10 and CW20 continual learning settings, as explicit task boundaries can often be conveniently determined using various methods, such as leveraging large models or human feedback on existing datasets or constructing new datasets with explicit task boundaries based on classical RL algorithms [1,2].
>
> Compared with task-aware continual learning, task boundary-agnostic continual learning requires an additional mechanism to detect whether a task change has occurred. Clearly, this demands extra task similarity measurement mechanisms for detection. Admittedly, this is currently a limitation of our approach. However, it is also one of the challenges we are actively addressing. We are confident that further progress will be reflected in our future research.
>
> > **[2/5] W2:** The SWA module contributes significantly even when task spaces are consistent. Could the authors elaborate on how SWA compares structurally and functionally to existing parameter-isolation methods such as PackNet? Is there a risk of capacity saturation or degradation in performance as tasks accumulate, especially in longer task sequences?
>
> Thank you for the valuable suggestions. We first discuss the differences between VQ-CD and PackNet.
> - Structurally, PackNet applies structured pruning to identify task-relevant weights and masks out the rest for later tasks, often involving iterative pruning-retraining cycles and reliance on weight magnitudes. In contrast, our SWA module adopts lightweight task-specific binary masks directly applied to convolution kernel weights, enabling efficient activation of relevant parameters during both training and inference. This approach avoids pruning heuristics and simplifies implementation within diffusion probabilistic models.
> - Functionally, PackNet was designed primarily for classification tasks with feedforward MLPs, while SWA of VQ-CD is designed to accommodate more sophisticated U-Net-based diffusion models. Besides, compared with PackNet, diffusion models' powerful expression and superior stability have been supported by recent studies [3].
>
> For the capacity saturation, it is an inherent challenge in all parameter-isolation methods, and we mitigate it in two ways: 1) Diffusion models are typically overparameterized, and early tasks often underutilize capacity, leaving room for subsequent tasks. 2) Mask pruning introduced in Appendix B.6 of the main body allows us to purposefully free up a portion of parameters for training new tasks. 3) For extremely long-horizon task sequences, we acknowledge the need for future work on adaptive mask reallocation, soft-mask learning, or latent reparameterization strategies to further improve scalability.
>
> > **[3/5] W3:** The authors mention that each task occupies a separate region in the quantized space. As the number of tasks increases, how does this design affect the size and efficiency of the codebook? Is there any mechanism to reuse or compress unused entries, or does the method assume a large fixed codebook with sufficient capacity?
>
> Thank you for your thoughtful suggestions. We try to address your concerns by discussing more details of our method.
>
> For different tasks, we maintain separate codebooks currently. If we think of the QSA module as a base station, these codebooks are like memory cards. When we need to test on task i, we load the corresponding codebook for task i from the saved checkpoints into the QSA module, enabling our model to perform space alignment for task i. When we encounter a new task j, we need to construct a new codebook for task j training and save it as a checkpoint for future use. Therefore, as the number of tasks increases, the memory overhead of QSA during inference does not increase with the addition of tasks, because QSA only needs to load the task-relevant codebook for evaluation.
>
> To further address potential codebook saturation or inefficiency in longer task sequences, we believe several extensions are promising directions: 1) Codebook LoRA strategy, which means new tasks can reuse several old codebook latents. 2) Codebook regularization based on usage frequency that encourages all tasks to share a single codebook, where new tasks during QSA training should minimize the use of latents frequently called by previous tasks.
>
> > **[4/5] W4:** In Figure 2, the performance of the multitask baseline, which is often considered the upper bound in continual learning, is unexpectedly poor. In Figure 3, VQ-CD even substantially outperforms multitask training. Could the authors provide insights or possible reasons for this?
>
> There are two main reasons that contribute to this result:
> - **Architecture differences:** The multitask baseline is implemented using the MLP architecture, which is often insufficient to model the complex state-action mappings across diverse tasks. In contrast, our VQ-CD method leverages a diffusion-based policy model, which has demonstrated superior expressiveness and stability in high-dimensional generative modeling [3]. This difference leads to the performance differences in the experiments.
> - **Representation ambiguity across tasks:** As shown in Table 2 of the main body, the QSA module maps task-specific state-action pairs into a shared aligned latent space, where the distances between different tasks' state- and action-aligned vectors are relatively small. This results in high semantic proximity across tasks. For Multitask training, all tasks' state- and action-aligned vectors are blended together, which makes it challenging to learn the mapping of a state-aligned vector to the correct action-aligned vector, leading to degraded performance. In contrast, VQ-CD separates alignment with QSA and generation via the conditional diffusion model, which enables more precise modeling and avoids such interference.
>
> > **[5/5] W5:** Given that VQ-CD builds a shared latent space across tasks, is the model able to generalize to unseen tasks with similar structure but new state/action configurations?
>
> Thank you for your forward-thinking insights.
> While our method currently focuses on CORL, we agree that the structure of our framework naturally supports generalization to new tasks with similar temporal or semantic structure. Given the shared latent space across tasks, the QSA module learns to project diverse task-specific state and action vectors into a unified discrete latent space. This design inherently promotes composability and reuse of latent codes for unseen tasks.
>
> While we have not explicitly evaluated zero-shot or few-shot generalization in this paper, the proposed framework can be extended to support such scenarios with appropriate adaptations. For example, a new task with different state and action spaces but similar temporal semantics can be encoded via the shared latent space. Then the decision can be produced by diffusion models followed by the QSA decoder.
>
> ### References
>
> [1] Continual World: A Robotic Benchmark For Continual Reinforcement Learning.
>
> [2] Continual offline reinforcement learning via diffusion-based dual generative replay.
>
> [3] Diffusion policies as an expressive policy class for offline reinforcement learning.
>
>
> Please do not hesitate to let us know if you have any further concerns.
>
> Sincerely,
>
> Authors of Paper 6316

---

> > ### Comment · Reviewer_Yyw3 · 2025-08-04
> >
> > Thank you for the rebuttal. It has addressed my concerns, and I will raise my score accordingly. I hope the authors will revise the manuscript to reflect the clarifications provided and correct any imprecise statements in the final version.

---

> > > ### Author Response · Authors · 2025-08-05
> > > **Appreciation to Reviewer Yyw3**
> > >
> > > Dear Reviewer Yyw3,
> > >
> > > We sincerely appreciate your willingness to reconsider your score based on our clarifications. We will carefully revise the manuscript to incorporate the explanations provided in the rebuttal and ensure that any imprecise statements are revised in the final version.
> > >
> > > Thanks again for your constructive comments and support!
> > >
> > > Best regards,
> > >
> > > Authors of Paper 6316

---

### Official Review · Reviewer_aMmx · 2025-07-02

**Clarity:** 2
**Significance:** 2
**Originality:** 3
**Rating:** 4
**Confidence:** 4

**Summary:**

This paper tackled continual offline RL via two components: quantized spaces alignment (QSA) and selective weights activation (SWA). QSA adopts ensemble vector quantized encoders to align state-action spaces and task-related decoders to recover original spaces. SWA selectively activates different weights according to task-related sparse masks.

**Questions:**

- The SWA and QSA modules seems to have nothing to do with diffusion models, and can be applied to any other network architectures. In what ways do the two modules match the utilization of diffusion models?

- The QSA is trained using Eq. (4). However, in experiments, the spaces are aligned using simple state-action padding. Any explanation on this inconsistency?

- It is hard to figure out how the encoder and decoder in QSA are trained. It seems to map different state-action spaces into a fixed-length latent space, and then decode the original space during inference. What is the intrinsic principle when designing the QSA module, and in what way QSA can extract common knowledge for tasks with different state-action spaces?

- The network masking strategy is trivial and heuristic. Also, the strategy is friendly to MLPs. What is the scalability when applied to transformers?

- In the arbitrary CL setting of Figure 3, are there only three tasks on the stream? Also, the performance does not degrade when switching to a new task, which is bit of counter-intuitive.

**Ethical Concerns:**

["NO or VERY MINOR ethics concerns only"]

**Final Justification:**

The authors provided further clarification and partially addressed my concerns. I increased my original score.

**Limitations:**

Yes

**Quality:**

2

**Strengths And Weaknesses:**

Strengths:
- The way of aligning state-action spaces is necessary and interesting.
- The comparison to multiple baselines is comprehensive.

Weaknesses:
- The motivation of using diffusion models is unclear and weak. The core modules of QSA and SWA seem applicable to any network architectures, not just for diffusion models.

- The method is too complicated, failing to yield an elegant formulation. The presentation is also too complicated and confusing in many parts (such as Fig. 1, too many components, failing to identify the critical parts).

- The SWA module involve many hand-designed heuristics, leading to the impression of insufficient contribution on the methodology side.

---

> ### Author Rebuttal · Authors · 2025-07-31
>
> Dear Reviewer aMmx,
>
> We appreciate the reviewer's efforts for reviewing and respond to your concerns below.
>
> ### [I]. Response to Weaknesses
>
> > **[1/3] W1:** The motivation of using diffusion models is unclear and weak. The core modules of QSA and SWA seem applicable to any network architectures, not just for diffusion models.
>
> Firstly, we would like to clarify why we chose diffusion models rather than other types of models. We select diffusion models as our foundational method primarily based on the following reasons.
> - **Natural support for multi-modal action distribution modeling.** Diffusion models can effectively avoid the limitations of traditional Gaussian models that cannot model multi-peak action distributions.
> - **Powerful model expressiveness.** Diffusion models support the modeling of complex data or trajectory distributions.
> - **Stable log-likelihood training.** Diffusion models can effectively prevent mode collapse and training instability issues.
> - **Competitive model performance.** Diffusion-based RL methods have also shown huge potential in many robotic control scenarios.
>
> Secondly, SWA is specifically designed for diffusion models with one-dimensional convolutional structures because diffusion models exhibit a different dependency pattern: every output of a given channel is influenced by all weights of the convolution kernel in that channel due to the weight-sharing nature of convolutions. Thus, traditional output neuron masking is not suitable for diffusion models with one-dimensional convolutional structures because we can not enable task-related parameters through masking certain output neurons. However, SWA applies masking directly on convolution kernel parameters, allowing us to selectively activate neurons in the convolution kernel to control the training of task-related parameters.
>
> Finally, QSA is applicable to any model. Different state and action spaces lead to significant distribution differences between tasks, making it impossible to use a single diffusion model for learning. QSA aligns these varying state and action spaces into the same latent space, enabling effective continual training under the same diffusion backbone.
>
> > **[2/3] W2:** The method is too complicated, failing to yield an elegant formulation. The presentation is also too complicated and confusing in many parts (such as Fig. 1, too many components, failing to identify the critical parts).
>
> Thanks for your helpful suggestions. We have drawn a simpler and more intuitive structural diagram to showcase the core parts of our method. Due to this year's updated rebuttal policy, we can not include new figures at this stage. But we will add the diagram and demonstrate the functions of QSA and SWA through intuitive examples in the revised version.
>
> > **[3/3] W3:** The SWA module involve many hand-designed heuristics, leading to the impression of insufficient contribution on the methodology side.
>
> We would like to clarify that the design of SWA is based on explicit motivation grounded in continual learning theory, particularly in the context of parameter isolation to mitigate forgetting. In diffusion models, previous parameter isolation strategies, such as output neuron masking, are improper for training due to the sophisticated 1D convolution structure. SWA addresses this by introducing task-specific binary masks over convolution kernels, enabling selective activation of parameters per task. Moreover, we explored other alternatives (e.g., sparse training) in Table 1 but found that the current SWA strikes the best balance between simplicity, effectiveness, and compatibility with diffusion models.
>
> **Table 1**: The comparison of time consumption per update between sparse and dense optimizers.
> |CL task|time consumption per update of dense training (s)|time consumption per update of dense training (s)|
> |-|-|-|
> |[Hopper-fr,Walker2d-fr,Halfcheetah-fr]|0.089|0.198|
> |[Hopper-mr,Walker2d-mr,Halfcheetah-mr]|0.096|0.197|
> |[Hopper-m,Walker2d-m,Halfcheetah-m]|0.089|0.195|
> |CW10|0.061|0.218|
>
> ### [II]. Response to Questions
>
> > **[1/5] Q1:** The SWA and QSA modules seems to have nothing to do with diffusion models, and can be applied to any other network architectures. In what ways do the two modules match the utilization of diffusion models?
>
> In **W1**, we detailed the importance of QSA and SWA for diffusion models. Please refer to **W1** for the detailed explanation.
>
> > **[2/5] Q2:** The QSA is trained using Eq. (4). However, in experiments, the spaces are aligned using simple state-action padding. Any explanation on this inconsistency?
>
> We want to clarify that we conducted experiments on the spaces aligned with QSA and displayed them in Figure 11 of the main body. We sincerely apologize for the confusion caused, and in the revised version, we will swap the positions of Figure 3 and Figure 11, first showing the QSA alignment results in the main text.
>
> > **[3/5] Q3:** It is hard to figure out how the encoder and decoder in QSA are trained. It seems to map different state-action spaces into a fixed-length latent space, and then decode the original space during inference. What is the intrinsic principle when designing the QSA module, and in what way QSA can extract common knowledge for tasks with different state-action spaces?
>
> The core principle behind QSA is to provide a shared representation basis, so that all tasks with different state and action spaces can be mapped into and decoded from the same quantized latent space. This provides a common ground for the inverse dynamics model, which can operate solely on the quantized latent states $s_{z_q}$ and learn the cross-task transition dynamics independent of the raw state $s$.
>
> This alignment is crucial for continual learning, as it reduces input heterogeneity and promotes shared policy reasoning. Our ablation study (shown in Figure 4 of the main body) and feature difference comparison (Please refer to Table 2 of the main body) further validate that QSA significantly improves cross-task representation consistency compared to other space alignment methods.
>
> > **[4/5] Q4:** The network masking strategy is trivial and heuristic. Also, the strategy is friendly to MLPs. What is the scalability when applied to transformers?
>
> We would like to emphasize that the Selective Weights Activation (SWA) module is carefully designed to address the unique challenges of diffusion models in continual learning, rather than being a trivial or ad hoc solution. In contrast to transformer-based methods, diffusion-based methods that are built on U-Net convolutional backbones are more suitable for modeling dense temporal state-action distributions. Diffusion models' powerful expression and superior stability have been supported by recent works [1]. The design of SWA is grounded in convolution-based diffusion models, enabling parameter isolation by introducing task-specific binary masks over convolution kernels, and shows potential in addressing various continual learning tasks.
>
> Regarding scalability to transformers: 1) While transformers may permit more granular activation control (e.g., attention head masking), they also introduce significant memory and computation overhead, especially in long-horizon RL. 2) Our SWA is architecture-agnostic in principle, and could be adapted to transformer blocks by masking feedforward or attention parameters.
>
> > **[5/5] Q5:** In the arbitrary CL setting of Figure 3, are there only three tsks on the stream? Also, the performance does not degrade when switching to a new task, which is bit of counter-intuitive.
>
> The experiments in Figure 3 contain three tasks, and the y-axis represents the average performance across these three tasks. This means that when the model is trained on the first task, the tested performance is divided by 3. Since SWA preserves knowledge of historical tasks through parameter isolation, when switching to a new task, the model's knowledge of historical tasks is preserved to the greatest extent possible, so performance hardly declines.
>
> ### References
>
> [1] Is Conditional Generative Modeling all you need for Decision-Making?
>
>
> Please do not hesitate to let us know if you have any further concerns.
>
> Sincerely,
>
> Authors of Paper 6316

---

> > ### Comment · Reviewer_aMmx · 2025-08-04
> > **Thanks for clarification**
> >
> > I would like to thank the authors to clarify my concerns. Some points have been addressed while the others are not completely addressed. The motivation is improved but not very convinced. The clarification to W2 is unclear. The implication of Table 1 is unclear and the same column title "time consumption per update of dense training (s)" is confusing.

---

> > > ### Author Response · Authors · 2025-08-04
> > > **Official Response to Reviewer aMmx**
> > >
> > > Dear Reviewer aMmx,
> > >
> > > We appreciate the reviewer's further feedback.
> > >
> > > > Some points have been addressed while the others are not completely addressed. The motivation is improved but not very convinced. The clarification to W2 is unclear. The implication of Table 1 is unclear and the same column title "time consumption per update of dense training (s)" is confusing.
> > >
> > > In response to your concerns regarding the motivation clarity, the explanation of W2, and the column title ambiguity in Table 1, we try to address your concerns by providing the following clarifications.
> > >
> > > **Clarification on the motivation.** We choose the diffusion model as the base framework for continual offline reinforcement learning due to several advantages: 1) its natural capacity to model multi-modal action distributions, 2) strong expressive power, 3) stable model training, and 4) empirically verified effectiveness in recent studies [1-4]. Building upon the diffusion model, we propose the SWA module to specifically mitigate catastrophic forgetting. This design is motivated by the limitations of existing parameter isolation strategies, e.g., output neuron masking, which are not compatible with diffusion models due to their reliance on all parameters in the 1D convolutional architectures. SWA overcomes this challenge by introducing task-specific binary masks applied directly to convolutional kernels, thereby enabling selective parameter activation for each task. Additionally, the QSA module extends the applicability of our method to continual learning scenarios by aligning inconsistent state-action spaces into a unified latent space.
> > >
> > > **Clarification on W2.** We fully appreciate your concern regarding the overall complexity of the proposed method and the presentation, particularly in Figure 1. In the revised version, we have redrawn a simplified schematic of the framework, explicitly highlighting the functional roles of QSA and SWA, while omitting low-level architectural details of QSA, SWA, and the diffusion model. We have also refined the corresponding textual descriptions of the workflow (shown below) to focus on the core mechanisms. Following your suggestions, we believe these revisions will significantly improve the clarity and accessibility of the overall framework.
> > >
> > > We provide the workflow for this picture below.
> > > - On the left, multiple tasks (e.g., Hopper, Walker2d, HalfCheetah, Ant) with heterogeneous state and action spaces are first processed by the QSA module. This module projects all tasks into a shared latent space with unified dimensions, enabling consistent training across tasks.
> > > - The middle part shows that the tasks are processed sequentially after alignment. As the system transitions from one task to the next, the SWA module is employed to isolate task-specific parameters using binary masks over the diffusion model's convolutional kernels. This mechanism helps preserve previously acquired knowledge while learning new tasks.
> > > - As shown in the right part, once all tasks are trained, the learned weights from each task are aggregated through the Weights Assembling process, resulting in a unified diffusion model capable of handling multiple tasks.
> > >
> > >
> > > **Clarification on Table 1.** We apologize for any confusion caused by the original column title. In the revised version of Table 1, we clarify that "dense optimizer" refers to using the standard Adam optimizer to update network parameters, while "sparse optimizer" refers to using a sparse variant of the Adam optimizer. The table is intended to demonstrate that the physical time cost of the training with "dense optimizer + weights assembling" is substantially lower than that of the sparse optimizer when training and extracting task-specific parameters. This efficiency consideration motivates our design of training with "dense optimizer + weights assembling".
> > >
> > > **Table 1**: The comparison of time consumption per update between dense and sparse optimizers.
> > > |Tasks|time consumption of per network updating with **dense optimizer** (s)|time consumption of per network updating with **sparse optimizer** (s)|
> > > |-|-|-|
> > > |[Hopper-fr,Walker2d-fr,Halfcheetah-fr]|0.089|0.198|
> > > |[Hopper-mr,Walker2d-mr,Halfcheetah-mr]|0.096|0.197|
> > > |[Hopper-m,Walker2d-m,Halfcheetah-m]|0.089|0.195|
> > > |CW10|0.061|0.218|
> > >
> > >
> > > ### References
> > >
> > > [1] Planning with diffusion for flexible behavior synthesis.
> > >
> > > [2] Diffusion policies as an expressive policy class for offline reinforcement learning.
> > >
> > > [3] Continual offline reinforcement learning via diffusion-based dual generative replay.
> > >
> > > [4] t-DGR: A trajectory-based deep generative replay method for continual learning in decision making.
> > >
> > >
> > > Thanks again for your review and suggestions.
> > > Please do not hesitate to let us know if you have any further concerns.
> > >
> > > Sincerely,
> > >
> > > Authors of Paper 6316

---

> > > > ### Comment · Reviewer_aMmx · 2025-08-06
> > > >
> > > > I would like to thank the authors' further clarification. I increase my score.
> > > >
> > > > The authors mentioned "we have redrawn a simplified schematic of the framework, explicitly highlighting the functional roles of QSA and SWA, while omitting low-level architectural details of QSA, SWA, and the diffusion model". Redrawing the framework and omitting details seem to not naturally reduce the complexity.

---

> > > > > ### Author Response · Authors · 2025-08-06
> > > > > **Appreciation to Reviewer aMmx**
> > > > >
> > > > > Dear Reviewer aMmx,
> > > > >
> > > > > Thanks for raising the score. We truly appreciate your constructive, insightful, and specific feedback. Your comments have been instrumental in helping us significantly improve the clarity and overall quality of our paper.
> > > > >
> > > > > To ensure full transparency, we will include all the clarification details mentioned in the rebuttal in the revised version of the manuscript. We believe this will strike a balance between clarity in presentation and completeness in explanation. We would like to clarify that the purpose of the simplified schematic of the framework is to help readers quickly grasp the core ideas without being overwhelmed by architectural details at first glance.
> > > > >
> > > > > Best regards,
> > > > >
> > > > > Authors of Paper 6316

---

### Official Review · Reviewer_YNrb · 2025-07-03

**Clarity:** 3
**Significance:** 3
**Originality:** 3
**Rating:** 4
**Confidence:** 4

**Summary:**

This paper tackles the problem of continual offline RL where action and observation space can be different among tasks. To this end, it proposed a method termed Vector-Quantized Continual Diffuser VQ-CD. VQ-CD comprises two parts, namely the quantization spaces alignment where vector quantization is used to align the different state and action spaces of different tasks, allowing continual learning in the same space. The second part is to use a unified diffusion model attached to by the inverse dynamic model to master all tasks by selectively activating different weights according to task-related sparse masks. The paper also provided experiments to demonstrates that VQ-CD can achieve SOTA performance on15 continual learning tasks (compared with 17 baselines).

**Questions:**

1.      The method developed in this work is for offline RL settings. I wonder how difficult or what are the factors that can enable this method to be applied to online RL settings. It might be better to provide a little more discussion regarding why the proposed method is a favorable solution for offline RL, rather than RL in general?
2.      On line 120, should q(\tau_s) be q(\tau_s^0)? Also for \tau_s^{1:K}, where k\in[1:K] is referred to as the diffusion step, whereas on line 163, \tau^_s^I, where $i$ indicates task index, are these two subscripts consistent with each other? On line 148, what is the expression for the stop gradient operation, sg(.)?
3.      The tasks considered in this work are mostly continuous control problems similar to each other. Will the proposed method work if tasks are of different types, for example after playing some video game for a few hours, you may consider to do some exercise like swimming or running, and then do homework, like reading comprehension or math?
4.      For table 1, is there a particular reason the continual task sequence is chosen to be 10-15-19-25. Will the results or conclusion be different if a different task sequence is chosen?

**Ethical Concerns:**

["NO or VERY MINOR ethics concerns only"]

**Final Justification:**

Given the quality of this paper and possible room for improvement, I maintain my rating.

**Limitations:**

yes

**Quality:**

3

**Strengths And Weaknesses:**

riginality.
Strength:
1.      Novelty: comparing to many previous works on continual learning where action space and observation space are the same and tasks are mainly distinguished through varied reward or dynamics. This works considers a different setting, where observation and action spaces are different.
2.       Comprehensive empirical evaluations are provided to demonstrate the superior performance of the proposed method over SOTA
Weaknesses:
1.      Most results are empirical, no theoretical analysis, such as convergence property, or sample/computation complexity are provided
2.      The evaluation tasks are limited to existing continuous control problems
3.      Although the proposed method works well empirically, it is unclear to me how this method can be applied to a general continue learning setting, like the one I provided in Question 3. If might be better if at the beginning of a paper, some intuitive examples can be provided to illustrate how the proposed method works, besides Figure 1.

---

> ### Author Rebuttal · Authors · 2025-07-31
>
> Dear Reviewer YNrb,
>
> We appreciate the reviewer's efforts for reviewing and respond to your concerns below.
>
> ### [I]. Response to Weaknesses
>
> > **[1/3] W1:** Most results are empirical, no theoretical analysis, such as convergence property, or sample/computation complexity are provided
>
> #### **[W1.1] Theoretical Analysis**
>
> We discuss theoretical analysis from two perspectives.
>
> **Convergence proof of QSA:** Inspired by the proof of VQ-VAE [1], under Robbins–Monro stepsizes and Lipschitz gradients, the expected gradient vanishes as the training iteration increases. In orther words, $\lim_{n\rightarrow\infty} \mathbb{E}[||\nabla_{\theta_e,\theta_d} \mathcal{L}_{QSA}||]$, where $n$ is the training interation. The proof contains three steps.
> - The encoder and decoder parameters are updated via projected stochastic gradient descent, ensuring that each step remains within a bounded feasible set and that the expected loss does not increase.
> - The vector quantization step follows an EM-style update, which guarantees non-increasing reconstruction loss.
> - Since the total loss is bounded below by zero and decreases in expectation, the update sequence converges, and the expected gradient norm tends to zero as training progresses.
>
> **Convergence proof of SWA:** According to the previous studies of diffusion model [2,3], the denoising score-matching loss ($\mathcal{L}(\theta)=\mathbb{E}[||\epsilon-\epsilon_{\theta}(\tau_s^k, k, b*\mathcal{C})||_2^2]$) is a lower bound on the negative log-likelihood. Under the Lipschitz constraint, the loss landscape is smooth, and optimization using Adam or SGD with diminishing learning rates satisfies the Robbins–Monro conditions. Thus, the training process converges to a stationary point of the objective.
>
> #### **[W1.2] Computation Complexity**
>
> Following your suggestions, we provide the computation complexity.
>
> We compare the computation complexity by comparing with diffusion-based methods, such as CuGRO, and transformer-based methods, such as L2M, and report the results in Table 1 and Table 2. The results show that, compared to the baselines, our method achieves lower time overhead and better performance with similar memory usage.
>
> **Table 1**: The computational cost of generation speed with different generation steps in D4RL [Hopper-m,Walker2d-m,Halfcheetah-m] tasks.
> ||base|VQ-CD|CuGRO|
> |-|-|-|-|
> |Time consumption|5.73|0.29|0.33|
> |speed-up ratio|1×|19.8×|17.4×|
> |score|-|45.4|27.6|
>
> **Table 2**: The comparison of GPU memory consumption. We control the hardware the same for all methods. The GPU is NVIDIA GeForce RTX 3090, and the CPU is Intel(R) Xeon(R) Gold 6230 @ 2.10GHz.
> |domain|CL task setting|Method|GPU memory overhead (GB)|Approximate physical training time consumption (h)|Performance|
> |-|-|-|-|-|-|
> |D4RL|[Hopper-m,Walker2d-m,Halfcheetah-m]|VQ-CD (Ours)|4.6|55|48.0|
> |D4RL|[Hopper-m,Walker2d-m,Halfcheetah-m]|L2M|6.7|92|13.4|
> |D4RL|[Hopper-m,Walker2d-m,Halfcheetah-m]|L2M-large|8.8|94|16.0|
>
> > **[2/3] W2:** The evaluation tasks are limited to existing continuous control problems
>
> We appreciate your comments. Though our current evaluation focuses on continuous control benchmarks, we would like to clarify that the chosen tasks already cover a wide spectrum of challenges:
> 1) **Task diversity:** Our experiments include multiple types of continual learning tasks across locomotion and robotic manipulation, with different state and action spaces and difficulty levels.
> 2) **Space heterogeneity:** In D4RL experiments, we test across tasks with heterogeneous state and action spaces, such as Hopper, Walker2d, and HalfCheetah, under various dataset qualities (e.g., medium and full-replay), which simulate realistic distribution shifts.
>
> More importantly, the core framework of VQ-CD is task-agnostic and modular, which makes it conceptually applicable to other types of tasks, such as visual RL tasks, as long as it is provided with appropriate encoder and decoder adaptations. We agree that evaluating on broader domains would further enhance generality, and we consider investigating these directions in future work.
>
> > **[3/3] W3:** Although the proposed method works well empirically, it is unclear to me how this method can be applied to a general continue learning setting, like the one I provided in Question 3. If might be better if at the beginning of a paper, some intuitive examples can be provided to illustrate how the proposed method works, besides Figure 1.
>
> Generally speaking, previous methods rarely consider continual learning on totally different RL tasks, such as playing video games followed by exercise [4,5]. Such high-level task heterogeneity involves dramatic shifts in not just state and action spaces, but also in task dynamics, goals, and modality. There are essential challenges that we need to resolve.
> - First, there are inconsistencies in state and action spaces between different tasks, which would prevent a single model from adapting to each task. QSA may provide an effective way to align the different state and action spaces.
> - Second, the reward function structures vary significantly between different tasks, which will affect the joint distribution modeling between returns and state sequences with diffusion models. Currently, reward normalization is a simple but effective method.
> - The task difference between different tasks is substantial, such as autonomous driving tasks and robot control tasks. Acquiring prior knowledge of tasks and designing task-specific action decoders for each task might be a direct and effective solution.
>
> We fully agree that intuitive examples in the introduction can largely improve accessibility for readers. In the revised version, we will incorporate a motivating example, e.g., learning to control different robots to master different skills, to concretely illustrate how VQ-CD aligns representations, learns knowledge, and reduces forgetting.
>
> ### [II]. Response to Questions
>
> > **[1/4] Q1:** It might be better to provide a little more discussion regarding why the proposed method is a favorable solution for offline RL, rather than RL in general?
>
> Our method is currently designed for the offline RL setting primarily due to the multiple-step sampling process inherent in diffusion models, which leads to significant computational latency during interaction in online RL. To enable online RL extensions in the future, several promising research lines can be explored:
> - Parallelized sampling techniques to reduce per-decision latency.
> - Advanced samplers (e.g., DDIM [3] and DPM-solver [4]) to accelerate generation.
> - Distilling the diffusion model into the consistency model [5] that can realize single-step generation.
>
> > **[2/4] Q2:** On line 120, should q(\tau_s) be q(\tau_s^0)? Also for \tau_s^{1:K}, where k\in[1:K] is referred to as the diffusion step, whereas on line 163, \tau^_s^I, where $i$ indicates task index, are these two subscripts consistent with each other? The explanation of sg(.)?
>
> 1. Your understanding is right, $q(\tau_s)$ and $q(\tau_s^0)$ are same each other. Usually, we omit the diffusion step subscript if it is 0.
> 2. In line 163, $i$ denotes the task index and the diffusion step supscript of $\tau^i_{s_{z_q}}$ is 0, thus we omit the diffusion step.
> 3. Stop Gradient refers to blocking the gradient propagation path of certain tensors to the loss function during backpropagation, ensuring that these tensors will not be updated in the current training process.
>
> > **[3/4] Q3:** Will the proposed method work if tasks are of different types, for example video game followed by some exercise like swimming or running, and then do homework, like reading comprehension or math?
>
> We sincerely thank the reviewer's advice. Please refer to **W3** for a detailed discussion.
>
> > **[4/4] Q4:** For table 1, is there a particular reason the continual task sequence is chosen to be 10-15-19-25. Will the results or conclusion be different if a different task sequence is chosen?
>
> We arbitrarily selected several tasks from Ant-dir to form a continual learning sequence. To verify that our method is independent of task selection, we also select other tasks to form task sequences, such as the task sequence 4-18-26-34-42-49 and longer sequences such as 4-9-18-22-26-34-42-49, and report the performance comparison with other baselines in Table 3.
> Furthermore, to verify that our method is independent of the sequence order of tasks, we also arbitrarily rearranged the order of tasks for experiments.
>
> Experimental results show that our method outperforms baselines across various arbitrarily selected task sequences, and achieves similar performance across them, indicating that our method's performance is independent of task selection and insensitive to the sequence order of tasks. Meanwhile, some baselines are more sensitive to the sequence order of tasks. For example, PackNet performs better when the task sequence is 4-26-18-34-42-49, which differs significantly from its performance on the 18-4-26-34-42-49.
>
> **Table 3**: The experiments of arbitrary task selections.
> |Task sequences|VQ-CD|CRIL|CuGRO|PackNet|DGR|t-DGR|Finetune|EWC|
> |-|-|-|-|-|-|-|-|-|
> |4-18-26-34-42-49|524.1|408.4|385.6|289.8|100.2|72.3|56.4|50.9|
> |4-9-18-22-26-34-42-49|523.6|328.1|293.9|241.3|105.0|88.6|84.3|84.9|
> |18-4-26-34-42-49|516.7|435.1|340.1|97.5|83.8|77.4|56.6|58.1|
> |49-4-34-18-26-42|518.7|403.8|275.7|136.6|187.3|186.5|278.3|278.5|
> |4-26-18-34-42-49|517.2|421.1|344.6|291.0|99.5|56.4|58.2|57.2|
>
> ### References
>
> [1] Neural discrete representation learning.
>
> [2] Denoising diffusion probabilistic models.
>
> [3] Denoising diffusion implicit models.
>
> [4] DPM-Solver: A Fast ODE Solver for Diffusion Probabilistic Model Sampling in Around 10 Steps.
>
> [5] Consistency Models.
>
>
> Please do not hesitate to let us know if you have any further concerns.
>
> Sincerely,
>
> Authors of Paper 6316

---

> > ### Comment · Reviewer_YNrb · 2025-08-08
> >
> > I appreciate the authors' effort on addressing my concerns and questions. I think this is a solid paper if it can be revised based on content in the rebuttal. The paper can be further improvement by considering more diverse tasks and image-based input (as suggested by another reviewer), hence I maintain my rating.

---

### Official Review · Reviewer_tiFq · 2025-07-03

**Clarity:** 3
**Significance:** 3
**Originality:** 3
**Rating:** 5
**Confidence:** 3

**Summary:**

The paper addresses the challenge of continual multi-task RL where an agent must learn multiple tasks without forgetting previously known knowledge. The authors propose a novel framework Vector-Quantized Continual Diffuser, named VQ-CD,
1. suitable for multi-tasks and a new continuing task  with a different space
2. A quantized space alignment (QSA) which maps the space to a quantized latent space  to align with any state/action space
3. Selective weights activation (SWA) diffuser module to reduce the knowledge forgetting by adjust the activation of the weights.

The evaluation is comprehensive and conducted across several benchmarks, including Ant-dir and Continual World, and D4RL.

**Questions:**

1. What motivated the architectural decision to separate state sequence modeling (diffusion model) from action prediction (inverse dynamics model) rather than employing a unified model for complete trajectories? Is this separation related to accommodating different action spaces across tasks?


2. Given the strong performance demonstrated within individual benchmarks, have you considered training across different benchmark environments simultaneously? What key challenges would you anticipate in such a cross-benchmark continual learning scenario?


3. How would VQ-CD perform when faced with dramatic shifts in state representation formats across tasks? For instance, what adaptations would be necessary to handle a transition from low-dimensional vector state spaces to high-dimensional image-based observations while maintaining knowledge transfer capabilities?


4. The VQ-CD framework integrates multiple technical components (quantized space alignment, selective weight activation, task-specific regularization). Which of these elements do you consider most critical to the method's performance gains?

5. The authors justify selecting parameter masking over output neuron masking based on purported difficulties in "distinguishing dependencies from weights to loss during backward propagation." However, modern automatic differentiation frameworks like PyTorch are specifically designed to handle such dependencies transparently. Could you clarify why dependency tracking poses a significant challenge in your architecture when automatic differentiation systems should resolve this issue inherently?

**Ethical Concerns:**

["NO or VERY MINOR ethics concerns only"]

**Final Justification:**

5 AC

**Limitations:**

1. High-Dimensional State Representation Evaluation? Testing on high-dimensional observation spaces, such as image-based environments, would be even better to see the power of the proposed method.


2. Despite empirical success, the framework lacks theoretical analysis for example, explaining why the quantized latent space representation achieves effective knowledge transfer between tasks

3. Computational Efficiency Concerns

**Paper Formatting Concerns:**

-

**Quality:**

3

**Strengths And Weaknesses:**

Strengths:
- A novel approach that combines the quantized latent space, powerful diffusion models, and several improvements to tackle the continual multi-task offline RL problem, which increasingly becoming more important
- Experimental results demonstrate remarkable performance advantages over existing methods across multiple benchmark environments
- Clear presentation with logical structure and comprehensive technical details enhances readability and reproducibility

Weakness:
- Computational efficiency and resource requirements are not adequately addressed in the current analysis

---

> ### Author Rebuttal · Authors · 2025-07-31
>
> Dear Reviewer tiFq,
>
> We appreciate the reviewer's efforts for reviewing and respond to your concerns below.
>
> ### [I]. Response to Weaknesses
>
> > **[1/1] W1:** Computational efficiency analysis
>
> We compare the computational cost, including generation time and memory consumption, with diffusion-based methods, such as CuGRO, and transformer-based methods, such as L2M, and report the results in Table 1 and Table 2. The results show that, compared to the baselines, our method achieves lower time overhead and better performance with similar memory usage.
>
> **Table 1**: The computational cost of generation speed with different generation steps in D4RL [Hopper-m,Walker2d-m,Halfcheetah-m] tasks.
> ||base|VQ-CD|CuGRO|
> |-|-|-|-|
> |Time consumption|5.73|0.29|0.33|
> |speed-up ratio|1×|19.8×|17.4×|
> |score|-|45.4|27.6|
>
> **Table 2**: The comparison of GPU memory consumption. We control the hardware the same for all methods. The GPU is NVIDIA GeForce RTX 3090, and the CPU is Intel(R) Xeon(R) Gold 6230 @ 2.10GHz.
> |domain|CL task setting|Method|GPU memory overhead (GB)|Approximate physical training time consumption (h)|Performance|
> |-|-|-|-|-|-|
> |D4RL|[Hopper-m,Walker2d-m,Halfcheetah-m]|VQ-CD (Ours)|4.6|55|48.0|
> |D4RL|[Hopper-m,Walker2d-m,Halfcheetah-m]|L2M|6.7|92|13.4|
> |D4RL|[Hopper-m,Walker2d-m,Halfcheetah-m]|L2M-large|8.8|94|16.0|
>
> ### [II]. Response to Questions
>
> > **[1/5] Q1:** The motivation of inverse dynamics.
>
> Following previous studies [1], inverse dynamics is introduced to produce actions based on the state sequence generated by the diffusion model. We choose to model the distribution of state sequence rather than state-action sequence on the basis of two reasons:
> - Usually, in many robotics control scenarios, the actions are often represented as joint torques, which are high-frequency and less smooth, making it hard to model and predict the action sequence.
> - The state is usually continuous in RL, but the mode of action is diverse, such as discrete and continuous. Modeling state sequences separately makes the diffusion-based model more generic to extensive RL scenarios.
>
> Using the diffusion model to model the state sequences and producing actions with the inverse dynamics are not related to accommodating different action spaces across tasks.
>
> To further investigate the benefit of producing actions with inverse dynamics rather than generating (s,a) together with diffusion models, we conduct the experiments of modeling state and action sequences together with diffusion models and only modeling state sequences with diffusion models. Table 3 results show that when using inverse dynamics, our method can achieve better performance compared with directly producing action with diffusion models.
>
> **Table 3**: The comparison of producing actions with the diffusion model and the inverse dynamics.
> |type|producing action with diffusion model|producing action with inverse dynamics|
> |-|-|-|
> |performance in Ant-dir 4-18-26-34-42-49|498.2|524.1|
> |performance in [Hopper-m,Walker2d-m,Halfcheetah-m]|39.5|45.4|
>
> > **[2/5] Q2:** Training across different benchmark environments. What key challenges?
>
> Generally speaking, previous methods rarely consider continual learning across benchmarks, because totally different RL tasks will bring several challenges [2,3]. We discuss them below and propose potential solutions.
> - First, there are inconsistencies in state and action spaces between different tasks, which would prevent a single model from adapting to each task. Fortunately, the quantized space alignment can be used to learn in a unified latent space.
> - The reward function structures vary significantly between different tasks, which will affect the joint distribution modeling between returns and state sequences with diffusion models. To address this issue, returns need to be normalized.
> - The task difference between different tasks is substantial, such as autonomous driving tasks and robot control tasks. Acquiring prior knowledge of tasks and designing task-specific action decoders for each task might be a direct and effective solution.
>
> Given the problems posed by these challenges, we will consider training across different benchmark environments simultaneously in our future research.
>
> > **[3/5] Q3:** Adaptation from low-dimensional vector state spaces to high-dimensional image-based observations
>
> Thank you for your comments.
> Usually, the applicability of continual learning across tasks with completely different state spaces is quite narrow, and previous studies rarely explore this area [3,4]. The purpose of continual learning is to progressively master new tasks by discovering common knowledge between tasks.
>
> There are several substantial challenges facing high-dimensional image-based observations, particularly in terms of representation alignment, codebook generalization, and action generation.
> - **For representation:** We can introduce a visual encoder, e.g., ViT, to extract compact latent representations from images before passing them into the diffusion models, thereby aligning the modalities.
> - **For codebook:** The QSA module can be modified to support modality-specific encoders while maintaining a shared codebook space or adopting modality-conditional quantization.
> - **For inverse dynamics:** If we use inverse dynamics to directly produce actions with image-like states, then we need to upgrade inverse dynamics to a vision-conditioned architecture.
>
> > **[4/5] Q4:** Discussion of technical components used in VQ-CD
>
> Quantized space alignment and selective weight activation each play important roles. Quantized space alignment is used to expand the application range of VQ-CD, while selective weight activation is used to reduce forgetting of historical tasks in continual learning. The advantages of these two modules are complementary, and performance gains rely on both modules simultaneously.
>
> > **[5/5] Q5:** Why dependency tracking poses a significant challenge?
>
> Our decision to prefer parameter masking over output neuron masking stems from the 1D convolutional network structure of the diffusion model.
>
> In classical MLP-based architectures, each output neuron is directly and exclusively associated with a subset of weights, making output masking an effective mechanism to selectively deactivate certain parts of the model. In contrast, 1D convolutional networks exhibit a different dependency pattern: every output of a given channel is influenced by all weights of the convolution kernel in that channel due to the weight-sharing nature of convolutions.
>
> Consequently, even if we apply output neuron masking at the activation level, the autograd engine of PyTorch will still compute gradients and update all convolutional weights associated with the masked outputs, as those weights contribute to other unmasked outputs. This deviates from the purpose of selective activation of task-specific parameters. Therefore, to achieve accurate control over which convolutional parameters are updated per task, we adopt parameter masking at the kernel level.
>
> ### [III]. Response to Limitations
>
> > **[1/3] L1:** High-Dimensional State Representation Evaluation?
>
> Thank you for your comments. Although our current work focuses on low-dimensional state spaces, the core design of VQ-CD, particularly the modular quantized space alignment (QSA), can be extended to high-dimensional modalities like images.
> - **For space alignment.** We can incorporate the visual encoder (e.g., ResNet or ViT) to extract latent features from images before quantization, enabling modality alignment without changing the core pipeline.
> - **For codebook generalization.** The QSA module already supports modular encoder-decoder pairs and scalable codebooks, allowing us to integrate image-based encoders and maintain aligned latent spaces across modalities.
> - **For action generation:** The inverse dynamics can be upgraded to a vision-conditioned architecture to handle transitions between visual observations.
>
> While image-based evaluation is beyond the current scope, we believe these extensions are feasible within our framework, and we will investigate them in future work.
>
> > **[2/3] L2:** Why the quantized latent space representation is effective
>
> We acknowledge that the current version lacks theoretical proof to demonstrate that space-aligned representations can effectively extract common knowledge between tasks. However, our method has a theoretical foundation for the representation with vector quantization [5,6]. In our paper, the QSA module maps tasks' states and actions into a shared discrete latent space using the codebook. This process enables structural alignment between otherwise heterogeneous observation spaces, making it easier for the diffusion model to generalize across tasks.
>
> About the convergence of QSA, we can prove that under Robbins–Monro stepsizes and Lipschitz gradients, the expected gradient of the QSA loss $\lim_{n\rightarrow\infty} \mathbb{E}[||\nabla_{\theta_e,\theta_d} \mathcal{L}_{QSA}||]$ vanishes as the training iteration increases, where $n$ is the training iteration [7].
>
> We will investigate further theoretical analysis in our future work. Besides, we will add the discussion of the limitations in the main body.
>
> > **[3/3] L3:** Computational Efficiency
>
> We sincerely appreciate your suggestions. Please refer to W1 for the experiments on computational efficiency.
>
> ### References
>
> [1] Is conditional generative modeling all you need for decision-making?
>
> [2] Continual world: A robotic benchmark for continual reinforcement learning.
>
> [3] Three types of incremental learning.
>
> [4] Continual Offline Reinforcement Learning via Diffusion-based Dual Generative Replay.
>
> [5] Theory and experiments on vector quantized autoencoders.
>
> [6] Vector quantization for adaptive state aggregation in reinforcement learning.
>
> [7] Neural discrete representation learning.
>
> Please do not hesitate to let us know if you have any further concerns.
>
> Sincerely,
>
> Authors of Paper 6316

---

### Decision · Program_Chairs · 2025-09-17

**Decision:**

Accept (poster)

**Comment:**

Within the continual offline learning framework, the paper proposes VQ-CD, which aligns different observation and action spaces via vector quantization and uses selective weight activation to preserve prior tasks. This approach will be quite useful to the community with strong results across different benchmarks. I recommend accepting this paper.

However, reviewers noted a lack of high-dimensional image settings. In addition, I also suggest clarifying for a broader audience the difference between offline and online continual learning and what aspect of the work is specific to offline learning or not, etc, that will help the continual learning community put this work into perspective. Also, please address the remaining issues noted by the reviewers such as analysis on computational efficiency and resource requirements, some of which are already discussed by the authors.